# Large Language Models for Automated Data Science: Introducing CAAFE for Context-Aware Automated Feature Engineering

**Noah Hollmann**
University of Freiburg
Charité Hospital Berlin
Prior Labs
noah.homa@gmail.com

**Samuel Müller**
University of Freiburg

Prior Labs
muellesa@cs.uni-freiburg.de

**Frank Hutter**
University of Freiburg

Prior Labs
fh@cs.uni-freiburg.de

## Abstract

As the field of automated machine learning (AutoML) advances, it becomes increasingly important to incorporate domain knowledge into these systems. We present an approach for doing so by harnessing the power of large language models (LLMs). Specifically, we introduce Context-Aware Automated Feature Engineering (CAAFE), a feature engineering method for tabular datasets that utilizes an LLM to iteratively generate additional semantically meaningful features for tabular datasets based on the description of the dataset. The method produces both Python code for creating new features and explanations for the utility of the generated features.

Despite being methodologically simple, CAAFE improves performance on 11 out of 14 datasets - boosting mean ROC AUC performance from 0.798 to 0.822 across all dataset - similar to the improvement achieved by using a random forest instead of logistic regression on our datasets.

Furthermore, CAAFE is interpretable by providing a textual explanation for each generated feature. CAAFE paves the way for more extensive semi-automation in data science tasks and emphasizes the significance of context-aware solutions that can extend the scope of AutoML systems to semantic AutoML. We release our code, a simple demo and a python package.

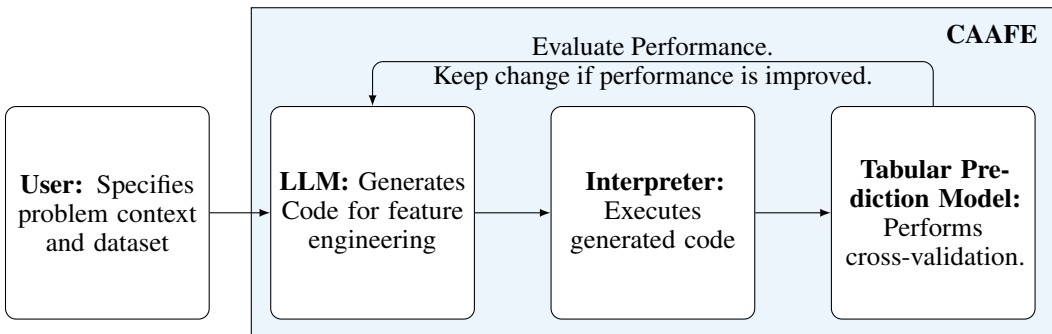

Figure 1: CAAFE accepts a dataset as well as user-specified context information and operates by iteratively proposing and evaluating feature engineering operations.

37th Conference on Neural Information Processing Systems (NeurIPS 2023).

# 1 Introduction

Automated machine learning (AutoML; e.g., Hutter et al. (2019)) is very effective at optimizing the machine learning (ML) part of the data science workflow, but existing systems leave tasks such as data engineering and integration of domain knowledge largely to human practitioners. However, model selection, training, and scoring only account for a small percentage of the time spent by data scientists (roughly 23% according to the "State of Data Science"(Anaconda, 2020)). Thus, the most time-consuming tasks, namely data engineering and data cleaning, are only supported to a very limited degree by AutoML tools, if at all.

While the traditional AutoML approach has been fruitful and appropriate given the technical capabilities of ML tools at the time, large language models (LLMs) may extend the reach of AutoML to cover more of data science and allow it to evolve towards *automated data science* (De Bie et al., 2022). LLMs encapsulate extensive domain knowledge that can be used to automate various data science tasks, including those that require contextual information. They are, however, not interpretable, or verifiable, and behave less consistently than classical ML algorithms. E.g., even the best LLMs still fail to count or perform simple calculations that are easily solved by classical methods (Hendrycks et al., 2021; OpenAI Community, 2021).

In this work, we propose an approach that combines the scalability and robustness of classical ML classifiers (e.g. random forests (Breiman, 2001)) with the vast domain knowledge embedded in LLMs, as visualized in Figure 2. We bridge the gap between LLMs and classical algorithms by using code as an interface between them: LLMs generate code that modifies input datasets, these modified datasets can then be processed by classical algorithms. Our proposed method, CAAFE, generates Python code that creates semantically meaningful features that improve the performance of downstream prediction tasks in an iterative fashion and with algorithmic feedback as shown in Figure 1. Furthermore, CAAFE generates a comment for each feature which explains the utility of generated feature. This allows interpretable AutoML, making it easier for the user to understand a solution, but also to modify and improve on it. Our approach combines the advantages of classical ML (robustness, predictability and a level of interpretability) and LLMs (domain-knowledge and creativity).

Automating the integration of domain-knowledge into the AutoML process has clear advantages that extend the scope of existing AutoML methods. These benefits include: i) Reducing the latency from data to trained models; ii) Reducing the cost of creating ML models; iii) Evaluating a more informed space of solutions than previously possible with AutoML, but a larger space than previously possible with manual approaches for integrating domain knowledge; and iv) Enhancing the robustness and reproducibility of solutions, as computer-generated solutions are more easily reproduced. CAAFE demonstrates the potential of LLMs for automating a broader range of data science tasks and highlights the emerging potential for creating more robust and context-aware AutoML tools.

# 2 Background

## 2.1 Large Language Models (LLMs)

LLMs are neural networks that are pre-trained on large quantities of raw text data to predict the next word in text documents. Recently, GPT-4 has been released as a powerful and publicly available LLM (OpenAI, 2023a). The architecture of GPT-4 is not oublished, it is likely based on a deep neural network that uses a transformer architecture (Vaswani et al., 2017), large-scale pre-training on a diverse corpus of text and fine-tuning using reinforcement learning from human feedback (RLHF) (Ziegler et al., 2019). It achieves state-of-the-art performance on various tasks, such as text generation, summarization, question answering and coding. One can adapt LLMs to a specific task without retraining by writing a prompt (Brown et al., 2020; Wei et al., 2021); the model parameters are frozen and the model performs in-context inference tasks based on a textual input that formulates the task and potentially contains examples.

**LLMs as Tabular Prediction Models** Hegselmann et al. (2023) recently showed how to use LLMs for tabular data prediction by applying them to a textual representation of these datasets. A prediction on an unseen sample then involves continuing the textual description of that sample on the target column. However, this method requires encoding the entire training dataset as a string and processing it using a transformer-based architecture, where the computational cost increases

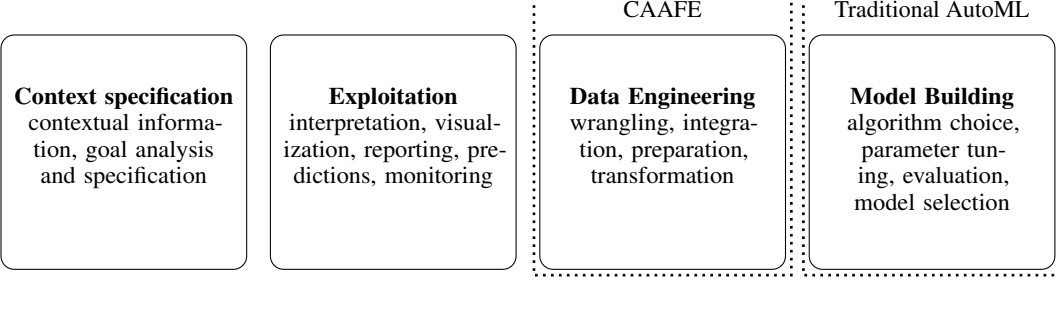

Figure 2: Data Science pipeline, inspired by De Bie et al. (2022). CAAFE allows for automation of semantic data engineering, while LLMs could provide even further automation: (1) Context specification is user driven (2) exploitation and data engineering can be automated through LLMs (3) model building can be automated by classical AutoML approaches.

quadratically with respect to $N \cdot M$, where $N$ denotes the number of samples and $M$ the number of features. Furthermore, the predictions generated by LLMs are not easily interpretable, and there is no assurance that the LLMs will produce consistent predictions, as these predictions depend directly on the complex and heterogeneous data used to train the models. So far, Hegselmann et al. (2023) found that their method yielded the best performance on tiny datasets with up to 8 samples, but was outperformed for larger data sets.

**LLMs for Data Wrangling** Narayan et al. (2022) demonstrated state-of-the-art results using LLMs for entity matching, error detection, and data imputation using prompting and manually tuning the LLMs. Vos et al. (2022) extended this technique by employing an improved prefix tuning technique. Both approaches generate and utilize the LLMs output for each individual data sample, executing a prompt for each row. This is in contrast to CAAFE, which uses code as an interface, making our work much more scalable and faster to execute, since one LLM query can be applied to all samples.

## 2.2 Feature Engineering

Feature engineering refers to the process of constructing suitable features from raw input data, which can lead to improved predictive performance. Given a dataset $D = (x_i, y_i)_{i=1}^{n}$, the goal is to find a function $\phi : \mathcal{X} \to \mathcal{X}'$ which maximizes the performance of $A(\phi(x_i), y_i)$ for some learning algorithm $A$. Common methods include numerical transformations, categorical encoding, clustering, group aggregation, and dimensionality reduction techniques, such as principal component analysis (Wold et al., 1987).

Deep learning methods are capable of learning suitable transformations from the raw input data making them more data-driven and making explicit feature engineering less critical, but only given a lot of data. Thus, appropriate feature engineering still improves the performance of classical and deep learning models, particularly for limited data, complex patterns, or model interpretability.

Various strategies for automated feature engineering have been explored in prior studies. Deep Feature Synthesis (DFS; Kanter & Veeramachaneni (2015)) integrates multiple tables for feature engineering by enumerating potential transformations on features and performing feature selection based on model performance. Cognito (Khurana et al., 2016) proposes a tree-like exploration of the feature space using handcrafted heuristic traversal strategies. AutoFeat (Horn et al., 2019) employs an iterative subsampling of features using beam search. Learning-based methods, such as LFE (Nargesian et al., 2017), utilize machine learning models to recommend beneficial transformations while other methods use reinforcement learning-based strategies (Khurana et al., 2018; Zhang et al., 2019). Despite these advancements, none of the existing methods can harness semantic information in an automated manner.

### 2.2.1 Incorporating Semantic Information

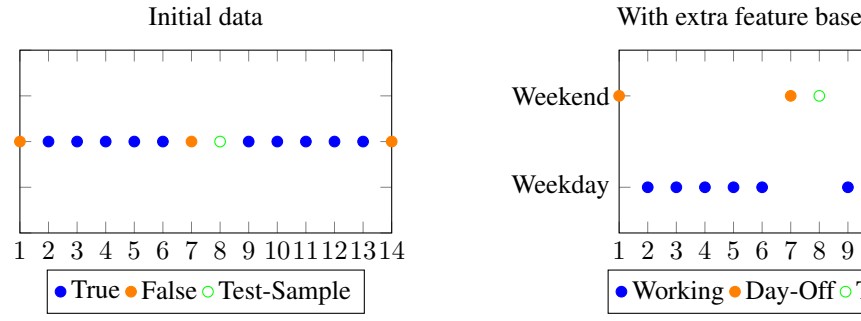

Figure 3: Contextual information can simplify a task immensely. On the left-hand side no contextual information is added to the plot, and it is hard to predict the label for the green query point. On the right-hand side contextual information is added and a useful additional feature (weekend or weekday) is derived from which a mapping from features to targets can be found.

The potential feature space, when considering the combinatorial number of transformations and combinations, is vast. Therefore, semantic information is useful, to serve as a prior for identifying useful features. By incorporating semantic and contextual information, feature engineering techniques can be limited to semantically meaningful features enhancing the performance by mitigating issues with multiple testing and computational complexity and boosting the interpretability of machine learning models. This strategy is naturally applied by human experts who leverage their domain-specific knowledge and insights. Figure 3 exemplifies the usefulness of contextual information.

## 3  Method

We present CAAFE, an approach that leverages large language models to incorporate domain knowledge into the feature engineering process, offering a promising direction for automating data science tasks while maintaining interpretability and performance.

Our method takes the training and validation datasets, $D_{train}$ and $D_{valid}$, as well as a description of the context of the training dataset and features as input. From this information CAAFE constructs a prompt, i.e. instructions to the LLM containing specifics of the dataset and the feature engineering task. Our method performs multiple iterations of feature alterations and evaluations on the validation dataset, as outlined in Figure 1. In each iteration, the LLM generates code, which is then executed on the current $D_{train}$ and $D_{valid}$ resulting in the transformed datasets $D'_{train}$ and $D'_{valid}$. We then use $D'_{train}$ to fit an ML-classifier and evaluate its performance $P'$ on $D'_{valid}$. If $P'$ exceeds the performance $P$ achieved by training on $D_{train}$ and evaluating on $D_{valid}$, the feature is kept and we set $D_{train} := D'_{train}$ and $D_{valid} := D'_{valid}$. Otherwise, the feature is rejected and $D_{train}$ and $D_{valid}$ remain unchanged. Figure 4 shows a shortened version of one such run on the Tic-Tac-Toe Endgame dataset.

**Prompting LLMs for Feature Engineering Code**   Here, we describe how CAAFE builds the prompt that is used to perform feature engineering. In this prompt, the LLM is instructed to create valuable features for a subsequent prediction task and to provide justifications for the added feature's utility. It is also instructed to drop unnecessary features, e.g. when their information is captured by other created features.

The prompt contains semantic and descriptive information about the dataset. Descriptive information, i.e. summary statistics, such as the percentage of missing values is based solely on the train split of the dataset. The prompt consists of the following data points:

 **A** A user-generated dataset description, that contains contextual information about the dataset (see Section 4 for details on dataset descriptions for our experiments)

 **B** Feature names adding contextual information and allowing the LLM to generate code to index features by their names

```
Dataset description: Tic-Tac-Toe Endgame database
This database encodes the complete set of possible board
    configurations at the end of tic-tac-toe games, where "x" is
    assumed to have played first. The target concept is "win for x" (i
    .e., true when "x" has one of 8 possible ways to create a "three-
    in-a-row").
```

```
# ('number-of-x-wins', 'Number of ways x can win on the board')
# Usefulness: Knowing the number of ways x can win on the board can be useful in
    predicting whether x has won the game or not.
# Input samples: 'top-left-square': [2, 2, 1], 'top-middle-square': [1, 2, 0], ...
df['number-of-x-wins'] = ((df['top-left-square']==1) & (df['top-middle-square']==1) & (df
    ['top-right-square']==1)).astype(int) + ((df['middle-left-square']==1) & (df['middle
    -middle-square']==1) & (df['middle-right-square']==1)).astype(int) [...]
```

```
Iteration 1
Performance before adding features ROC 0.888, ACC 0.700.
Performance after adding features ROC 0.987, ACC 0.980.
Improvement ROC 0.099, ACC 0.280. Code was executed and changes to df
    retained.
```

```
# ('number-of-o-wins', 'Number of ways o can win on the board')
# Usefulness: Knowing the number of ways o can win on the board can be useful in
    predicting whether o has won the game or not.
# Input samples: 'top-left-square': [2, 2, 1], 'top-middle-square': [1, 2, 0], ...
df['number-of-o-wins'] = ((df['top-left-square']==2) & (df['top-middle-square']==2) & (df
    ['top-right-square']==2)).astype(int) + ((df['middle-left-square']==2) & (df['middle
    -middle-square']==2) & (df['middle-right-square']==2)).astype(int) [...]
```

```
Iteration 2
Performance before adding features ROC 0.987, ACC 0.980.
Performance after adding features ROC 1.000, ACC 1.000.
Improvement ROC 0.013, ACC 0.020. Code was executed and changes to df
    retained.
```

Figure 4: Exemplary run of CAAFE on the Tic-Tac-Toe Endgame dataset. User generated input is shown in blue, ML-classifier generated data shown in red and LLM generated code is shown with syntax highlighting. The generated code contains a comment per generated feature that follows a template provided in our prompt (Feature name, description of usefulness, features used in the generated code and sample values of these features). In this run, CAAFE improves the ROC AUC on the validation dataset from 0.888 to 1.0 in two feature engineering iterations.

**C** Data types (e.g. float, int, category, string) - this adds information on how to handle a feature in the generated code

**D** Percentage of missing values - missing values are an additional challenge for code generation

**E** 10 random rows from the dataset - this provides information on the feature scale, encoding, etc.

Additionally, the prompt provides a template for the expected form of the generated code and explanations. Adding a template when prompting is a common technique to improve the quality of responses (OpenAI, 2023b). We use Chain-of-thought instructions – instructing a series of intermediate reasoning steps –, another effective technique for prompting (Wei et al., 2023). The prompt includes an example of one such Chain-of-thought for the code generation of one feature: first providing the high-level meaning and usefulness of the generated feature, providing the names of features used to generate it, retrieving sample values it would need to accept and finally writing a line of code. We provide the complete prompt in Figure 5 in the appendix.

If the execution of a code block raises an error, this error is passed to the LLM for the next code generation iteration. We observe that using this technique CAAFE recovered from all errors in our experiments. One such example can be found in Table 3.

**Technical Setup** The data is stored in a Pandas dataframe (Wes McKinney, 2010), which is preloaded into memory for code execution. The generated Python code is executed in an environment where the training and validation data frame is preloaded. The performance is measured on the current dataset with ten random validation splits $D_{valid}$ and the respective transformed datasets $D'_{valid}$ with the mean change of accuracy and ROC AUC used to determine if the changes of a code block are kept, i.e. when the average of both is greater than 0. We use OpenAI's GPT-4 and GPT-3.5 as LLMs (OpenAI, 2023a) in CAAFE. We perform ten feature engineering iterations and TabPFN (Hollmann et al., 2022) in the iterative evaluation of code blocks.

The automatic execution of AI-generated code carries inherent risks, such as misuse by malicious actors or unintended consequences from AI systems operating outside of controlled environments. Our approach is informed by previous studies on AI code generation and cybersecurity (Rohlf, 2023; Crockett, 2023). We parse the syntax of the generated python code and use a whitelist of operations that are allowed for execution. Thus operations such as imports, arbitrary function calls and others are excluded. This does not provide full security, however, e.g. does not exclude operations that can lead to infinite loops and excessive resource usage such as loops and list comprehensions.

## 4  Experimental Setup

**Setup of Downstream-Classifiers** We evaluate our method with Logistic Regression, Random Forests (Breiman, 2001) and TabPFN (Hollmann et al., 2022) for the final evaluation while using TabPFN to evaluate the performance of added features. We impute missing values with the mean, one-hot or ordinal encoded categorical inputs, normalized features and passed categorical feature indicators, where necessary, using the setup of Hollmann et al. (2022) [1].

**Setup of Automated Feature Engineering Methods** We also evaluate popular context-agnostic feature engineering libraries Deep Feature Synthesis (DFS; Kanter & Veeramachaneni, 2015) and AutoFeat (Horn et al., 2019)[2]. We evaluate DFS and AutoFeat alone and in combination with CAAFE. When combined, CAAFE is applied first and the context-agnostic AutoFE method subsequently. For DFS we use the primitives "add_numeric" and "multiply_numeric", and default settings otherwise. For TabPFN, DFS generates more features than TabPFN accepts (the maximum number of features is 100) in some cases. In these cases, we set the performance to the performance without feature engineering. For AutoFeat, we use one feature engineering step and default settings otherwise.

**Evaluating LLMs on Tabular Data** The LLM's training data originates from the web, potentially including datasets and related notebooks. GPT-4 and GPT-3.5 have a knowledge cutoff in September 2021, i.e., almost all of its training data originated from before this date. Thus, an evaluation on established benchmarks can be biased since a textual description of these benchmarks might have been used in the training of the LLM.

We use two categories of datasets for our evaluation: (1) widely recognized datasets from OpenML released before September 2021, that could potentially be part of the LLMs training corpus and (2) lesser known datasets from Kaggle released after September 2021 and only accessible after accepting an agreement and thus harder to access by web crawlers.

From OpenML (Vanschoren et al., 2013; Feurer et al.), we use small datasets that have descriptive feature names (i.e. we do not include any datasets with numbered feature names). Datasets on OpenML contain a task description that we provide as user context to our method. When datasets are perfectly solvable with TabPFN alone (i.e. reaches ROC AUC of 1.0) we reduce the training set size for that dataset, marked in Table 1. We focus on small datasets with up to 2 000 samples in total, because feature engineering is most important and significant for smaller datasets.

We describe the collection and preprocessing of datasets in detail in Appendix G.1.

---

[1] https://github.com/automl/TabPFN/blob/main/tabpfn/scripts/tabular_baselines.py
[2] https://github.com/alteryx/featuretools, https://github.com/cod3licious/autofeat

Table 1: ROC AUC OVO results using TabPFN. ± indicates the standard deviation across 5 splits. [R] indicates datasets where reduced data was used because TabPFN had 100% accuracy by default, see Appendix G.1.

| | TabPFN | | |
| | No Feat. Eng. | CAAFE (GPT-3.5) | CAAFE (GPT-4) |
|---|---|---|---|
| airlines | **0.6211** ±.04 | 0.619 ±.04 | 0.6203 ±.04 |
| balance-scale [R] | 0.8444 ±.29 | 0.844 ±.31 | **0.882** ±.26 |
| breast-w [R] | 0.9783 ±.02 | **0.9809** ±.02 | **0.9809** ±.02 |
| cmc | 0.7375 ±.02 | 0.7383 ±.02 | **0.7393** ±.02 |
| credit-g | 0.7824 ±.03 | 0.7824 ±.03 | **0.7832** ±.03 |
| diabetes | 0.8427 ±.03 | **0.8434** ±.03 | 0.8425 ±.03 |
| eucalyptus | **0.9319** ±.01 | 0.9317 ±.01 | **0.9319** ±.00 |
| jungle_chess.. | 0.9334 ±.01 | 0.9361 ±.01 | **0.9453** ±.01 |
| pc1 | 0.9035 ±.01 | 0.9087 ±.02 | **0.9093** ±.01 |
| tic-tac-toe [R] | 0.6989 ±.08 | 0.6989 ±.08 | **0.9536** ±.06 |
| $\langle Kaggle \rangle$ health-insurance | 0.5745 ±.02 | 0.5745 ±.02 | **0.5748** ±.02 |
| $\langle Kaggle \rangle$ pharyngitis | 0.6976 ±.03 | 0.6976 ±.03 | **0.7078** ±.04 |
| $\langle Kaggle \rangle$ kidney-stone | 0.7883 ±.04 | 0.7873 ±.04 | **0.7903** ±.04 |
| $\langle Kaggle \rangle$ spaceship-titanic | 0.838 ±.02 | 0.8383 ±.02 | **0.8405** ±.02 |

**Evaluation Protocol**   For each dataset, we evaluate 5 repetitions, each with a different random seed and train- and test split to reduce the variance stemming from these splits (Bouthillier et al., 2021). We split into 50% train and 50% test samples and all methods used the same splits.

# 5   Results

In this section we showcase the results of our method in three different ways. First, we show that CAAFE can improve the performance of a state-of-the-art classifier. Next, we show how CAAFE interacts with traditional automatic feature engineering methods and conclude with examples of the features that CAAFE creates.

Table 2: Mean ROC AUC and average rank (ROC AUC) per downstream classification method and feature extension method. Best AutoFE method per base classifer is shown in bold. The features generated by CAAFE are chosen with TabPFN as classifier. Rank are calculated across all classifiers and feature engineering methods. FETCH was too computationally expensive to compute for all base classifiers in the rebuttal. Each seed and dataset takes up to 24 hours and has to be evaluated for each base classifer independently. Thus, we use features computed for logistic regression for all other classifiers.

| | | | | Baselines | | | CAAFE | |
| | | No FE | DFS | AutoFeat | FETCH | OpenFE | GPT-3.5 | GPT-4 |
|---|---|---|---|---|---|---|---|---|
| Log. Reg. | Mean | 0.749 | 0.764 | 0.754 | 0.76 | 0.757 | 0.763 | **0.769** |
| | Mean Rank | 27.4 | **23.6** | 26.2 | 25.2 | 25 | 24.8 | 24.3 |
| Random Forest | Mean | 0.782 | 0.783 | 0.783 | 0.785 | 0.785 | 0.79 | **0.803** |
| | Mean Rank | 23.4 | 22.1 | 21.8 | 23.5 | 22.3 | 23.1 | **19.9** |
| ASKL2 | Mean | 0.807 | 0.801 | 0.808 | 0.807 | 0.806 | 0.815 | **0.818** |
| | Mean Rank | 12.2 | 12.9 | 12.6 | 13.4 | 13.5 | **10.9** | 11.6 |
| Autogluon | Mean | 0.796 | 0.799 | 0.797 | 0.787 | 0.798 | 0.803 | **0.812** |
| | Mean Rank | 17.6 | 15.4 | 16.4 | 17.6 | 16.6 | 15.8 | **14.1** |
| TabPFN | Mean | 0.798 | 0.791 | 0.796 | 0.796 | 0.798 | 0.806 | **0.822** |
| | Mean Rank | 13.9 | 15 | 14.8 | 16.5 | 13.9 | 12.9 | **9.78** |

**Performance of CAAFE**   CAAFE can improve our strongest classifier, TabPFN, substantially. If it is used with GPT-4, we improve average ROC AUC performance from 0.798 to 0.822, as shown in Table 2, and enhance the performance for 11/14 datasets. On the evaluated datasets, this improvement

Table 3: Examples of common strategies employed by CAAFE for feature extension. The full code and comments are automatically generated based on the user-provided dataset descriptions. CAAFE combines features, creates ordinal versions of numerical features through binning, performs string transformations, removes superfluous features, and even recovers from errors when generating invalid code.

| Description | Generated code |
|---|---|
| **Combination**
Example from the Kaggle Kidney Stone dataset. | ```python
# Usefulness: Fever and rhinorrhea are two of the most common
#     symptoms of respiratory infections, including GAS pharyngitis.
#     This feature captures their co-occurrence.
# Input samples: 'temperature': [38.0, 39.0, 39.5], 'rhinorrhea':
#     [0.0, 0.0, 0.0]
df['fever_and_rhinorrhea'] = ((df['temperature'] >= 38.0) & (df['
    rhinorrhea'] > 0)).astype(int)
``` |
| **Binning**
Example from the Kaggle Spaceship Titanic dataset. | ```python
# Feature: AgeGroup (categorizes passengers into age groups)
# Usefulness: Different age groups might have different likelihoods
#     of being transported.
# Input samples: 'Age': [30.0, 0.0, 37.0]
bins = [0, 12, 18, 35, 60, 100]
labels = ['Child', 'Teen', 'YoungAdult', 'Adult', 'Senior']
df['AgeGroup'] = pd.cut(df['Age'], bins=bins, labels=labels)
df['AgeGroup'] = df['AgeGroup'].astype('category')
``` |
| **String transformation**
Example from the Kaggle Spaceship Titanic dataset. | ```python
# Feature: Deck
# Usefulness: The deck information can help identify patterns in the
#     location of cabins associated with transported passengers.
# Input samples: 'Cabin': ['F/356/S', 'G/86/P', 'C/37/P']
df['Deck'] = df['Cabin'].apply(lambda x: x[0] if isinstance(x, str)
    else 'Unknown')

# Feature: CabinSide
# Usefulness: The side of the cabin can help identify patterns in
#     the location of cabins associated with transported passengers.
# Input samples: 'Cabin': ['F/356/S', 'G/86/P', 'C/37/P']
df['CabinSide'] = df['Cabin'].apply(lambda x: x.split('/')[-1] if
    isinstance(x, str) else 'Unknown')
``` |
| **Removing features**
Example from the Balance Scale dataset. | ```python
# Drop original columns
# Explanation: The original columns 'left-weight', 'left-distance',
#     'right-weight', and 'right-distance' can be dropped as the
#     newly created columns 'left_moment', 'right_moment', and '
#     moment_difference' capture the relevant information for
#     predicting the 'class'.
df.drop(columns=['left-weight', 'left-distance', 'right-weight', '
    right-distance'], inplace=True)
# No more additional columns needed
``` |
| **Error Recovery**
Example from the Kaggle Health Insurance dataset. The code execution failed because missing values were not considered in the dataframe. In the next iteration code is suggested that can handle missing values. | ```python
# Feature: Age_difference
# Usefulness: Age difference between upper and lower age can be
#     useful in determining the likelihood of a person applying for a
#     policy.
# Input samples: 'Upper_Age': ['C33', 'C2', 'C3'], 'Lower_Age': ['
#     Owned', 'Rented', 'Rented']
df['Age_difference'] = df['Upper_Age'].str.extract('(\d+)').astype(
    int) - df['Lower_Age'].str.extract('(\d+)').astype(int)

Feedback: failed with error: cannot convert float NaN to integer.

# Feature: Age_difference
# Usefulness: Age difference between upper and lower age can be
#     useful in determining the likelihood of a person applying for a
#     policy.
# Input samples: 'Upper_Age': ['C33', 'C2', 'C3'], 'Lower_Age': ['
#     Owned', 'Rented', 'Rented']
df['Age_difference'] = df['Upper_Age'].str.extract('(\d+)').astype(
    float).fillna(0) - df['Lower_Age'].str.extract('(\d+)').astype(
    float).fillna(0)
``` |

is similar (71%) to the average improvement achieved by using a random forest (AUC 0.783) instead of logistic regression (AUC 0.749). We can see that CAAFE even improves performance for all of the new datasets from Kaggle. If we use CAAFE with GPT-3.5 only, we can see that it performs clearly worse than with GPT-4, and only improves performance on 6/14 datasets.

There is great variability in the improvement size depending on whether (1) a problem is amenable to feature engineering, i.e. is there a mapping of features that explains the data better and that can be expressed through simple code; and (2) the quality of the dataset description (e.g., the balance-scale dataset contains an accurate description of how the dataset was constructed) Per dataset performance can be found in Table 1. CAAFE takes 4:43 minutes to run on each dataset, 90% of the time is spent on the LLM's code generation and 10% on the evaluation of the generated features. In Appendix F we plot the performance, time and cost of CAAFE across feature engineering iterations, showing the tradeoff between these parameters. For the 14 datasets, 5 splits and 10 CAAFE iterations, CAAFE generates 52 faulty features (7.4%) in the generation stage, from which it recovers (see Figure 3).

**Incorporating Classical AutoFE Methods**    Classical AutoFE methods can readily be combined with our method, one simply runs CAAFE first and then lets a classical AutoFE method find further feature extensions, as we did in Table 2. For less powerful downstream classifiers, namely Logistic Regression and Random Forests, we observe that applying AutoFE additionally to CAAFE improves performance further. The AutoML method TabPFN on the other hand is not improved by applying the evaluated AutoFE methods. This discrepancy might stem from the larger hypothesis space (complexity) of TabPFN, it can get all necessary information from the data directly. For all combinations of classifiers and additional AutoFE methods, we can see that CAAFE improves performance on average.

# 6   Conclusion

Our study presents a novel approach to integrating domain knowledge into the AutoML process through Context-Aware Automated Feature Engineering (CAAFE). By leveraging the power of large language models, CAAFE automates feature engineering for tabular datasets, generating semantically meaningful features and explanations of their utility. Our evaluation demonstrates the effectiveness of this approach, which complements existing automated feature engineering and AutoML methods.

This work emphasizes the importance of context-aware solutions in achieving robust outcomes. We expect that LLMs will also be useful for automating other aspects of the data science pipeline, such as data collection, processing, model building, and deployment. As large language models continue to improve, it is expected that the effectiveness of CAAFE will also increase.

Dataset descriptions play a critical role in our method; however, in our study, they were derived solely from web-crawled text associated with public datasets. If users were to provide more accurate and detailed descriptions, the effectiveness of our approach could be significantly improved.

However, our current approach has some limitations. Handling datasets with a large number of features can lead to very large prompts, which can be challenging for LLMs to process effectively. The testing procedure for adding features is not based on statistical tests, and could be improved using techniques of previous feature engineering works. LLMs, at times, exhibit a phenomenon known as "hallucinations.", where models produce inaccurate or invented information. Within CAAFE, this might result in the generation of features and associated explanations that appear significant and are logically presented, even though they may not be grounded in reality. Such behavior can be problematic, especially when individuals place trust in these systems for essential decision-making or research tasks. Finally, the usage of LLMs in automated data analysis comes with a set of societal and ethical challenges. Please see Section B for a discussion on the broader impact and ethical considerations.

Future research may explore prompt tuning, fine-tuning language models, and automatically incorporating domain-knowledge into models in other ways. Also, there may lie greater value in the interaction of human users with such automated methods, also termed human-in-the-loop AutoML (Lee & Macke, 2020), where human and algorithm interact continuously. This would be particularly easy with a setup similar to CAAFE, as the input and output of the LLM are interpretable and easily modified by experts.

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

# A  Acknowledgements

GPT-4 (OpenAI, 2023a) was used in the following ways: to help us iterate on LaTeX formatting and diagram plotting; for text summarization; and as a copyediting tool; for rephrasing proposals.

# B  Broader Impact Statement

## B.1  Social Impact of Automation

The broader implications of our research may contribute to the automation of data science tasks, potentially displacing workers in the field. However, CAAFE crucially depends on the users inputs for feature generation and processing and provides an example of human-in-the-loop AutoML. The automation of routine tasks could free up data scientists to focus on higher-level problem-solving and decision-making activities. It is essential for stakeholders to be aware of these potential consequences, and to consider strategies for workforce education and adaptation to ensure a smooth transition as AI technologies continue to evolve.

## B.2  Replication of Biases

AI algorithms have been observed to replicate and perpetuate biases observed in their training data distribution. CAAFE leverages GPT-4, which has been trained on web crawled data that contains existing social biases and generated features may be biased on these biases. An example study of such biases in web-crawled data was done by Prabhu & Birhane (2020). When data that contains demographic information or other data that can potentially be used to discriminate against groups, we advise not to use CAAFE or to proceed with great caution, double checking the generated features.

We would like to mention the best practice studies on fairness for AutoML by Weerts et al. (2023) and on bias in the use of LLMs by Talboy & Fuller (2023).

Weerts et al. (2023) give recommendations for what AutoML developers can do to improve fairness-conscious AutoML, and we follow the three most emphasized closely. i) We clearly emphasize the limitations and biases of our method. i) We emphasize the 'Usefulness' comment that our prompt asks the LLM to write into our UI to "help users identify sources of fairness-related harm", as Weerts et al. (2023) write. ii) We execute code only after asking the user for explicit confirmation that it does not contain dangerous actions or biases; these kinds of design decisions are called "seam-full design" by (Weerts et al., 2023).

(Talboy & Fuller, 2023) give recommendations on how to best use LLMs. Their recommendation, which we as interface developers can more easily follow, is the one they emphasize the most. Namely, that "LLMs [...] should be used as decision support tools, not final decision makers. We follow this closely by not allowing the LLM to execute code without the user's explicit consent.

The choice of benchmarks can also introduce dangerous biases in machine learning methods, as there are only a few that dominate the landscape (Raji et al., 2021). We address this problem by using datasets from two different sources (Kaggle and OpenML). In addition, we use multiple datasets from each source.

**How Can Biases Arise in CAAFE?**  Since we are using a trained model (the LLM) to define the training setup (features used) of a downstream classifier trained on potentially biased data, we can have bias at three levels: i) at the level of the LLM, and ii) at the level of the generated features, ii) at the level of the downstream classifier.

The level of the downstream classifier is affected by the previous two levels, but is not directly affected by CAAFE, so we do not discuss it in detail. Here, as in any data science application, the user must be careful to use unbiased data. The way we set up CAAFE has some robustness against biases in the first two steps, which are controlled by CAAFE:

i) We use GPT-4, where great care has been taken to mitigate biases – still it does contain biases in its generations. We add a statement that choosing a language model that is trained to mitigate biases is a crucial step for the user to tackle this problem.

ii) Feature engineering operations by the LLM are only retained if they lead to improvements in cross-validation, due to the way our method is set up (we generate code and verify that it improves performance at each step). This gives us a slightly better defense against bias than using the LLM outputs directly, since biases that are present in the LLM but not in the data are discarded.

However, since the biases of the LLM and the dataset are most likely not independent, there is still a risk that the two will be aligned. To illustrate this risk and make users aware of potential problems, we include an illustrative Example (see end of this reply).

**Simple Example**   We have built a fake dataset that has only one feature, a person's name, and as output we want to predict whether the person is a doctor or a nurse. The description of the dataset used in our challenge is

```
Doctor-or-Nurse is a dataset that asks to predict whether a person is a doctor or a nurse
     just based on their name.
'''
The intentionally biased sample of the dataset shown to the LLM in the prompt is
'''
Columns in 'df' (true feature dtypes listed here, categoricals encoded as int):
Name (object): NaN-freq [0.0%], Samples ['Anna', 'Jack', 'Frank', 'Laura', 'Travis', '
     Sophia']
is_doctor (int): NaN-freq [0.0%], Samples [0, 1, 1, 0, 1, 0]
```

In all cases in this sample a name commonly associated with male gender is associated with a doctor. Additionally, we only used female names ending in an "a".

We can see that GPT-4 tends to suggest an attribute that uses the typically gender-associated ending of the name to classify, i.e. it generates the following output

```
# Feature: end_with_a
# Usefulness: Names ending with "a" might be more common in females. Assuming there might
     be a gender bias in the doctor or nurse profession, this could provide some useful
     information.
# Input samples: ['Anna', 'Jack', 'Frank']
df['end_with_a'] = df['Name'].str.endswith('a').astype(int)
```

Listing 1: CAAFE output

We can see that our prompt actually makes it very easy to detect the bias of the LLM in this case, since the 'Usefulness' comment the LLM is asked to provide, at least in this non-cherry-picked example, just gives away its bias.

### B.3   AI Model Interpretability

As the adoption of advanced AI methods grows, it becomes increasingly important to comprehend and interpret their results. Our approach aims to enhance interpretability by providing clear explanations of model outputs and generating simple code, thus making the automated feature engineering process more transparent.

### B.4   Risk of increasing AI capabilities

We do not believe this research affects the general capabilities of LLMs but rather demonstrates their application. As such we estimate our work does not contribute to the risk of increasing AI capabilities.

## C   Reproducibility

**Code release**   In an effort to ensure reproducibility, we release code to reproduce our experiments at https://github.com/automl/CAAFE We release a minimal demo at a simple demo.

**Availability of datasets** All datasets used in our experiments are freely available at OpenML.org (Van-schoren et al., 2014) or at kaggle.com, with downloading procedures included in the submission.

# D   Full LLM Prompt

Figure 5 shows the full prompt for one examplary dataset. The generated prompts are in our repository: https://github.com/automl/CAAFE/tree/main/data/generated_code.

```
The dataframe 'df' is loaded and in memory. Columns are also named attributes.
Description of the dataset in 'df' (column dtypes might be inaccurate):
"**Tic-Tac-Toe Endgame database**
This database encodes the complete set of possible board configurations at the end of tic-tac-
    toe games, where "x" is assumed to have played first. The target concept is "win for x"
    (i.e., true when "x" has one of 8 possible ways to create a "three-in-a-row"). "

Columns in 'df' (true feature dtypes listed here, categoricals encoded as int):
top-left-square (int32): NaN-freq [0.0%], Samples [2, 2, 2, 2, 2, 2, 0, 1, 1, 2]
top-middle-square (int32): NaN-freq [0.0%], Samples [0, 0, 1, 1, 1, 2, 0, 2, 2, 2]
top-right-square (int32): NaN-freq [0.0%], Samples [1, 0, 1, 2, 1, 1, 1, 0, 2, 1]
middle-left-square (int32): NaN-freq [0.0%], Samples [1, 0, 2, 1, 2, 0, 0, 2, 1, 2]
middle-middle-square (int32): NaN-freq [0.0%], Samples [0, 2, 2, 1, 2, 1, 1, 1, 2, 1]
middle-right-square (int32): NaN-freq [0.0%], Samples [1, 1, 2, 2, 2, 2, 0, 0, 0, 0]
bottom-left-square (int32): NaN-freq [0.0%], Samples [2, 1, 1, 0, 0, 1, 2, 0, 1, 1]
bottom-middle-square (int32): NaN-freq [0.0%], Samples [2, 0, 0, 0, 1, 2, 2, 2, 1, 0]
bottom-right-square (int32): NaN-freq [0.0%], Samples [2, 2, 0, 2, 0, 1, 2, 1, 2, 0]
Class (category): NaN-freq [0.0%], Samples [1.0, 1.0, 1.0, 1.0, 1.0, 0.0, 1.0, 0.0, 0.0, 0.0]

This code was written by an expert datascientist working to improve predictions. It is a
    snippet of code that adds new columns to the dataset.
Number of samples (rows) in training dataset: 71

This code generates additional columns that are useful for a downstream classification
    algorithm (such as XGBoost) predicting "Class".
Additional columns add new semantic information, that is they use real world knowledge on the
    dataset. They can e.g. be feature combinations, transformations, aggregations where the
    new column is a function of the existing columns.
The scale of columns and offset does not matter. Make sure all used columns exist. Follow the
    above description of columns closely and consider the datatypes and meanings of classes.
This code also drops columns, if these may be redundant and hurt the predictive performance of
    the downstream classifier (Feature selection). Dropping columns may help as the chance
    of overfitting is lower, especially if the dataset is small.
The classifier will be trained on the dataset with the generated columns and evaluated on a
    holdout set. The evaluation metric is accuracy. The best performing code will be selected
    .
Added columns can be used in other codeblocks, dropped columns are not available anymore.

Code formatting for each added column:
'''python
# (Feature name and description)
# Usefulness: (Description why this adds useful real world knowledge to classify "Class"
    according to dataset description and attributes.)
# Input samples: (Three samples of the columns used in the following code, e.g. 'top-left-
    square': [2, 2, 2], 'top-middle-square': [0, 0, 1], ...)
(Some pandas code using top-left-square', 'top-middle-square', ... to add a new column for
    each row in df)
'''end

Code formatting for dropping columns:
'''python
# Explanation why the column XX is dropped
df.drop(columns=['XX'], inplace=True)
'''end

Each codeblock generates exactly one useful column and can drop unused columns (Feature
    selection).
Each codeblock ends with '''end and starts with "'''python"
Codeblock:
```

Figure 5: Full LLM Prompt for the CMC dataset. The generated code will be the reply to this prompt.

# E    Additional Results

## E.1    Semantic Blinding

Semantic information, i.e. the context of the dataset and its columns, is crucial and can only be captured through laborious human work or our novel approach of using LLMs - this is the core of our approach. To further verify and quantify this claim, we perform an experiment where the context of the dataset is left out (i.e. feature names and dataset description are not given to the LLM). We find a strong drop in performance from an average AUROC of 0.822 to 0.8 over all datasets for GPT-4.

Table 4: CAAFE with semantic and without "Semantic Blinding". For "Semantic Blinding" feature names and the dataset description is concealed to CAAFE. Mean ROC AUC and average rank (ROC AUC) per downstream classification method and feature extension method is shown. Best performing AutoFE method per classifier is shown in bild. The features generated by CAAFE are chosen with TabPFN as classifier. Ranks are calculated across all classifiers and feature engineering methods.

|  |  | No FE | GPT-3.5 | | GPT-4 | |
|  |  |  | Semantic Blinding | Default | Semantic Blinding | Default |
|---|---|---|---|---|---|---|
| Log. Reg. | Mean | 0.749 | 0.754 | 0.763 | 0.749 | **0.769** |
|  | Mean Rank | 19.6 | 19.1 | 17.8 | 19.3 | **17.4** |
| Random Forest | Mean | 0.782 | 0.789 | 0.79 | 0.783 | **0.803** |
|  | Mean Rank | 17.2 | 16.5 | 16.8 | 16.5 | **14.6** |
| ASKL2 | Mean | 0.807 | 0.812 | 0.815 | 0.809 | **0.818** |
|  | Mean Rank | 9.17 | 9.29 | **8.16** | 8.88 | 8.59 |
| Autogluon | Mean | 0.796 | 0.803 | 0.803 | 0.801 | **0.812** |
|  | Mean Rank | 12.9 | 12.2 | 11.6 | 12.7 | **10.3** |
| TabPFN | Mean | 0.798 | 0.807 | 0.806 | 0.8 | **0.822** |
|  | Mean Rank | 10.5 | 9.33 | 9.71 | 9.33 | **7.59** |

## E.2    Per Dataset Results

Table 5: ROC AUC OVO results per dataset and downstream classification method. CAAFE optimized for strong performance on TabPFN.

|  | AutoFE-Base | | CAAFE | | AutoFE-Base + CAAFE | |
|  | AutoFeat | DFS | GPT-3.5 | GPT-4 | GPT-4 + AF | GPT-4 + DFS |
|---|---|---|---|---|---|---|
| airlines | **0.6211** ±.04 | 0.6076 ±.04 | 0.595 ±.04 | 0.619 ±.04 | 0.6203 ±.04 | 0.602 ±.04 | 0.5966 ±.04 |
| balance-scale | 0.8444 ±.29 | 0.8438 ±.30 | 0.8428 ±.31 | 0.844 ±.31 | **0.882** ±.26 | 0.8812 ±.27 | 0.8773 ±.27 |
| breast-w | 0.9783 ±.02 | 0.9713 ±.03 | 0.9783 ±.02 | **0.9809** ±.02 | **0.9809** ±.02 | 0.9713 ±.03 | **0.9809** ±.02 |
| cmc | 0.7375 ±.02 | 0.7384 ±.02 | 0.7349 ±.02 | 0.7383 ±.02 | **0.7393** ±.02 | 0.7386 ±.02 | 0.7362 ±.02 |
| credit-g | 0.7824 ±.03 | 0.7819 ±.03 | 0.7824 ±.03 | 0.7824 ±.03 | 0.7832 ±.03 | **0.784** ±.03 | 0.7824 ±.03 |
| diabetes | 0.8427 ±.03 | 0.8414 ±.03 | 0.8417 ±.03 | **0.8434** ±.03 | 0.8425 ±.03 | 0.8432 ±.03 | 0.8382 ±.03 |
| eucalyptus | 0.9319 ±.01 | 0.9321 ±.01 | 0.9319 ±.01 | 0.9317 ±.01 | 0.9319 ±.00 | **0.9323** ±.01 | 0.9319 ±.01 |
| jungle_chess.. | 0.9334 ±.01 | 0.9197 ±.01 | 0.9284 ±.01 | 0.9361 ±.01 | 0.9453 ±.01 | **0.9535** ±.01 | 0.94 ±.01 |
| $\langle Kaggle \rangle$ health-insurance | 0.5745 ±.02 | **0.5805** ±.03 | 0.5753 ±.02 | 0.5745 ±.02 | 0.5748 ±.02 | 0.5777 ±.03 | 0.5782 ±.03 |
| $\langle Kaggle \rangle$ pharyngitis | 0.6976 ±.03 | 0.6976 ±.03 | 0.6976 ±.03 | 0.6976 ±.03 | **0.7078** ±.04 | 0.7073 ±.04 | 0.6976 ±.03 |
| $\langle Kaggle \rangle$ kidney-stone | 0.7883 ±.04 | 0.7856 ±.04 | 0.7929 ±.04 | 0.7873 ±.04 | 0.7903 ±.04 | 0.7875 ±.04 | **0.7967** ±.03 |
| $\langle Kaggle \rangle$ spaceship-titanic | 0.838 ±.02 | 0.8486 ±.02 | 0.8443 ±.02 | 0.8383 ±.02 | 0.8405 ±.02 | **0.853** ±.02 | 0.8486 ±.02 |
| pc1 | 0.9035 ±.01 | 0.9046 ±.01 | 0.9035 ±.01 | 0.9087 ±.02 | **0.9093** ±.01 | 0.908 ±.01 | 0.9035 ±.01 |
| tic-tac-toe | 0.6989 ±.08 | 0.6989 ±.08 | 0.6291 ±.10 | 0.6989 ±.08 | **0.9536** ±.06 | **0.9536** ±.06 | 0.938 ±.06 |

## E.3    Generated Prompts and Code

You can find the generated prompts and the respective LLM generated code in our repository: https://github.com/cafeautomatedfeatures/CAFE/tree/main/data/generated_code.

# F  Compute

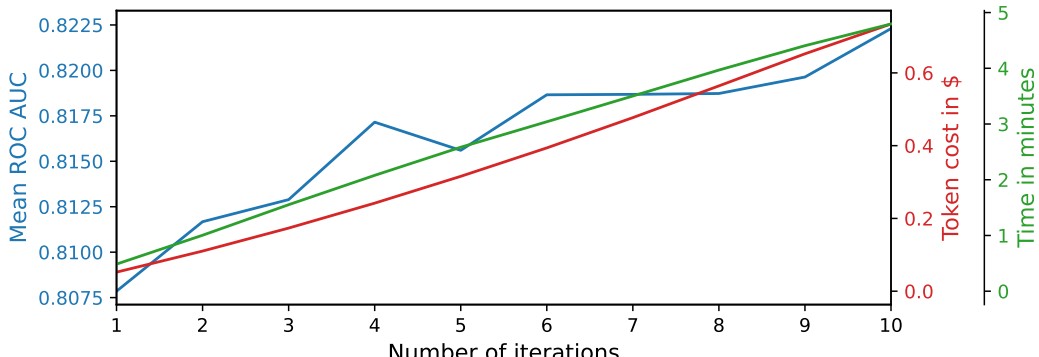

Figure 6: Mean ROC AUC OVO, inference cost for GPT and time spent with an increasing number of feature generation runs.

Figure F illustrates the increasing performance but also cost and time spent for more feature engineering iterations. Prediction for LLMs is done per token and so the generation of code takes dominates the 4:43 minutes evaluation time of CAAFE on average per dataset. For GPT-3.5 this time is reduce to about 1/4. Also for GPT-3.5 the cost is reduced to 1/10 as of the writing of this paper. For the evaluation of TabPFN we use one Nvidia RTX 2080 Ti as well as 8 Intel(R) Xeon(R) Gold 6242 CPU @ 2.80GHz CPU cores.

# G  Datasets

| Name | # Features | # Samples | # Classes | OpenML ID / Kaggle Name |
|---|---|---|---|---|
| balance-scale | 4 | 125 | 3 | 11 |
| breast-w | 9 | 69 | 2 | 15 |
| cmc | 9 | 1473 | 3 | 23 |
| credit-g | 20 | 1000 | 2 | 31 |
| diabetes | 8 | 768 | 2 | 37 |
| tic-tac-toe | 9 | 95 | 2 | 50 |
| eucalyptus | 19 | 736 | 5 | 188 |
| pc1 | 21 | 1109 | 2 | 1068 |
| airlines | 7 | 2000 | 2 | 1169 |
| jungle_chess_2pcs_raw_endgame_complete | 6 | 2000 | 3 | 41027 |
| pharyngitis | 19 | 512 | 2 | *pharyngitis* |
| health-insurance | 13 | 2000 | 2 | *health-insurance-lead-prediction-raw-data* |
| spaceship-titanic | 13 | 2000 | 2 | *spaceship-titanic* |
| kidney-stone | 7 | 414 | 2 | *playground-series-s3e12* |

Table 6: Test datasets used for the evaluation. See Section 4 for a description of the datasets used.

## G.1  Dataset Collection and Preprocessing

**OpenML datasets**  We use small datasets from OpenML (Vanschoren et al., 2013; Feurer et al.) that have descriptive feature names (i.e. we do not include any datasets with numbered feature names). Datasets on OpenML contain a task description that we provide as user context to our method and that we clean from redundant information for feature engineering, such as author names or release history. While some descriptions are very informative, other descriptions contain much less information. We remove datasets with more than 20 features, since the prompt length rises linearly with the number

of features and exceeds the permissible 8,192 tokens that standard GPT-4 can accept. We show all datasets we used in Table 6 in Appendix G. When datasets are perfectly solvable with TabPFN alone (i.e. reaches ROC AUC of 1.0) we reduce the training set size for that dataset to 10% or 20% of the original dataset size. This is the case for the datasets "balance-scale" (20%), "breast-w" (10%) and "tic-tac-toe" (10%). We focus on small datasets with up to $2\,000$ samples in total, because feature engineering is most important and significant for smaller datasets.

**Kaggle datasets**    We additionally evaluate CAAFE on $4$ datasets from Kaggle that were released after the knowledge cutoff of our LLM Model. These datasets contain string features as well. String features allow for more complex feature transformations, such as separating Names into First and Last Names, which allows grouping families. We drop rows that contain missing values for our evaluations. Details of these datasets can also be found in Table 6 in Appendix G.

### G.2 Dataset Descriptions

The dataset descriptions used were crawled from the respective datasource. For OpenML prompts uninformative information such as the source or reference papers were removed. Figures 20 show the parsed dataset descriptions used for each dataset.

```
**Balance Scale Weight & Distance Database**
This data set was generated to model psychological experimental results.  Each example is
    classified as having the balance scale tip to the right, tip to the left, or be
    balanced. The attributes are the left weight, the left distance, the right weight,
    and the right distance. The correct way to find the class is the greater of (left-
    distance * left-weight) and (right-distance * right-weight). If they are equal, it
    is balanced.

 Attribute description
The attributes are the left weight, the left distance, the right weight, and the right
    distance.
```

Figure 7: Dataset description for balance-scale.

```
**Breast Cancer Wisconsin (Original) Data Set.** Features are computed from a digitized
    image of a fine needle aspirate (FNA) of a breast mass. They describe
    characteristics of the cell nuclei present in the image. The target feature records
    the prognosis (malignant or benign).
```

Figure 8: Dataset description for breast-w.

```
   4. Relevant Information:
      This dataset is a subset of the 1987 National Indonesia Contraceptive
      Prevalence Survey. The samples are married women who were either not
      pregnant or do not know if they were at the time of interview. The
      problem is to predict the current contraceptive method choice
      (no use, long-term methods, or short-term methods) of a woman based
      on her demographic and socio-economic characteristics.

   7. Attribute Information:

      1. Wife's age                 (numerical)
      2. Wife's education           (categorical)      1=low, 2, 3, 4=high
      3. Husband's education        (categorical)      1=low, 2, 3, 4=high
      4. Number of children ever born (numerical)
      5. Wife's religion            (binary)           0=Non-Islam, 1=Islam
      6. Wife's now working?        (binary)           0=Yes, 1=No
      7. Husband's occupation       (categorical)      1, 2, 3, 4
      8. Standard-of-living index   (categorical)      1=low, 2, 3, 4=high
      9. Media exposure             (binary)           0=Good, 1=Not good
      10. Contraceptive method used (class attribute)  1=No-use
                                                       2=Long-term
                                                       3=Short-term
```

Figure 9: Dataset description for cmc.

```
**German Credit dataset**
This dataset classifies people described by a set of attributes as good or bad credit
    risks.

This dataset comes with a cost matrix:
'''
Good  Bad (predicted)
Good   0    1 (actual)
Bad    5    0
'''

It is worse to class a customer as good when they are bad (5), than it is to class a
    customer as bad when they are good (1).

 Attribute description

1. Status of existing checking account, in Deutsche Mark.
2. Duration in months
3. Credit history (credits taken, paid back duly, delays, critical accounts)
4. Purpose of the credit (car, television,...)
5. Credit amount
6. Status of savings account/bonds, in Deutsche Mark.
7. Present employment, in number of years.
8. Installment rate in percentage of disposable income
9. Personal status (married, single,...) and sex
10. Other debtors / guarantors
11. Present residence since X years
12. Property (e.g. real estate)
13. Age in years
14. Other installment plans (banks, stores)
15. Housing (rent, own,...)
16. Number of existing credits at this bank
17. Job
18. Number of people being liable to provide maintenance for
19. Telephone (yes,no)
20. Foreign worker (yes,no)
```

Figure 10: Dataset description for credit-g.

```
4. Relevant Information:
      Several constraints were placed on the selection of these instances from
      a larger database.  In particular, all patients here are females at
      least 21 years old of Pima Indian heritage.  ADAP is an adaptive learning
      routine that generates and executes digital analogs of perceptron-like
      devices.  It is a unique algorithm; see the paper for details.

7. For Each Attribute: (all numeric-valued)
   1. Number of times pregnant
   2. Plasma glucose concentration a 2 hours in an oral glucose tolerance test
   3. Diastolic blood pressure (mm Hg)
   4. Triceps skin fold thickness (mm)
   5. 2-Hour serum insulin (mu U/ml)
   6. Body mass index (weight in kg/(height in m)^2)
   7. Diabetes pedigree function
   8. Age (years)
   9. Class variable (0 or 1)

Relabeled values in attribute 'class'
   From: 0                     To: tested_negative
   From: 1                     To: tested_positive
```

Figure 11: Dataset description for diabetes.

```
**Tic-Tac-Toe Endgame database**
This database encodes the complete set of possible board configurations at the end of tic
    -tac-toe games, where "x" is assumed to have played first.  The target concept is "
    win for x" (i.e., true when "x" has one of 8 possible ways to create a "three-in-a-
    row").
```

Figure 12: Dataset description for tic-tac-toe.

```
**Eucalyptus Soil Conservation**
The objective was to determine which seedlots in a species are best for soil conservation
     in seasonally dry hill country. Determination is found by measurement of height,
     diameter by height, survival, and other contributing factors.

It is important to note that eucalypt trial methods changed over time; earlier trials
     included mostly 15 - 30cm tall seedling grown in peat plots and the later trials
     have included mostly three replications of eight trees grown. This change may
     contribute to less significant results.

Experimental data recording procedures which require noting include:
 - instances with no data recorded due to experimental recording procedures
   require that the absence of a species from one replicate at a site was
   treated as a missing value, but if absent from two or more replicates at a
   site the species was excluded from the site's analyses.
 - missing data for survival, vigour, insect resistance, stem form, crown form
   and utility especially for the data recorded at the Morea Station; this
   could indicate the death of species in these areas or a lack in collection
   of data.

 Attribute Information

  1.   Abbrev - site abbreviation - enumerated
  2.   Rep - site rep - integer
  3.   Locality - site locality in the North Island - enumerated
  4.   Map_Ref - map location in the North Island - enumerated
  5.   Latitude - latitude approximation - enumerated
  6.   Altitude - altitude approximation - integer
  7.   Rainfall - rainfall (mm pa) - integer
  8.   Frosts - frosts (deg. c) - integer
  9.   Year - year of planting - integer
  10.  Sp - species code - enumerated
  11.  PMCno - seedlot number - integer
  12.  DBH - best diameter base height (cm) - real
  13.  Ht - height (m) - real
  14.  Surv - survival - integer
  15.  Vig - vigour - real
  16.  Ins_res - insect resistance - real
  17.  Stem_Fm - stem form - real
  18.  Crown_Fm - crown form - real
  19.  Brnch_Fm - branch form - real
  Class:
  20.  Utility - utility rating - enumerated

 Relevant papers

Bulluch B. T., (1992) Eucalyptus Species Selection for Soil Conservation in Seasonally
     Dry Hill Country - Twelfth Year Assessment  New Zealand Journal of Forestry Science
     21(1): 10 - 31 (1991)

Kirsten Thomson and Robert J. McQueen (1996) Machine Learning Applied to Fourteen
     Agricultural Datasets. University of Waikato Research Report
https://www.cs.waikato.ac.nz/ml/publications/1996/Thomson-McQueen-96.pdf + the original
     publication:
```

Figure 13: Dataset description for eucalyptus.

```
Binarized version of the original data set (see version 1). The multi-class target
     feature is converted to a two-class nominal target feature by re-labeling the
     majority class as positive ('P') and all others as negative ('N'). Originally
     converted by Quan Sun.
```

Figure 14: Dataset description for wine.

```
**PC1 Software defect prediction**
One of the NASA Metrics Data Program defect data sets. Data from flight software for
    earth orbiting satellite. Data comes from McCabe and Halstead features extractors of
     source code.  These features were defined in the 70s in an attempt to objectively
    characterize code features that are associated with software quality.

 Attribute Information

1. loc             : numeric % McCabe's line count of code
2. v(g)            : numeric % McCabe "cyclomatic complexity"
3. ev(g)           : numeric % McCabe "essential complexity"
4. iv(g)           : numeric % McCabe "design complexity"
5. n               : numeric % Halstead total operators + operands
6. v               : numeric % Halstead "volume"
7. l               : numeric % Halstead "program length"
8. d               : numeric % Halstead "difficulty"
9. i               : numeric % Halstead "intelligence"
10. e              : numeric % Halstead "effort"
11. b              : numeric % Halstead
12. t              : numeric % Halstead's time estimator
13. lOCode         : numeric % Halstead's line count
14. lOComment      : numeric % Halstead's count of lines of comments
15. lOBlank        : numeric % Halstead's count of blank lines
16. lOCodeAndComment: numeric
17. uniq_Op        : numeric % unique operators
18. uniq_Opnd      : numeric % unique operands
19. total_Op       : numeric % total operators
20. total_Opnd     : numeric % total operands
21. branchCount    : numeric % of the flow graph
22. branchCount    : numeric % of the flow graph
23. defects        : {false,true} % module has/has not one or more reported defects

 Relevant papers

- Shepperd, M. and Qinbao Song and Zhongbin Sun and Mair, C. (2013)
Data Quality: Some Comments on the NASA Software Defect Datasets, IEEE Transactions on
    Software Engineering, 39.

- Tim Menzies and Justin S. Di Stefano (2004) How Good is Your Blind Spot Sampling Policy
    ? 2004 IEEE Conference on High Assurance
Software Engineering.

- T. Menzies and J. DiStefano and A. Orrego and R. Chapman (2004) Assessing Predictors of
    Software Defects", Workshop on Predictive Software Models, Chicago
```

Figure 15: Dataset description for pc1.

```
Airlines Dataset Inspired in the regression dataset from Elena Ikonomovska. The task is
    to predict whether a given flight will be delayed, given the information of the
    scheduled departure.
```

Figure 16: Dataset description for airlines.

```
    Description

This dataset is part of a collection datasets based on the game "Jungle Chess" (a.k.a.
    Dou Shou Qi). For a description of the rules, please refer to the paper (link
    attached). The paper also contains a description of various constructed features. As
     the tablebases are a disjoint set of several tablebases based on which (two) pieces
     are on the board, we have uploaded all tablebases that have explicit different
     content:

* Rat vs Rat
* Rat vs Panther
* Rat vs. Lion
* Rat vs. Elephant
* Panther vs. Lion
* Panther vs. Elephant
* Tiger vs. Lion
* Lion vs. Lion
* Lion vs. Elephant
* Elephant vs. Elephant
* Complete (Combination of the above)
* RAW Complete (Combination of the above, containing for both pieces just the rank, file
    and strength information). This dataset contains a similar classification problem as
    , e.g., the King and Rook vs. King problem and is suitable for classification tasks.

(Note that this dataset is one of the above mentioned datasets). Additionally, note that
    several subproblems are very similar. Having seen a given positions from one of the
    tablebases arguably gives a lot of information about the outcome of the same
    position in the other tablebases.

J. N. van Rijn and J. K. Vis, Endgame Analysis of Dou Shou Qi. ICGA Journal 37:2,
    120--124, 2014. ArXiv link: https://arxiv.org/abs/1604.07312
```

Figure 17: Dataset description for jungle_chess_2pcs_raw_endgame_complete.

```
For the data and objective, it is evident that this is a Binary Classification Problem
    data in the Tabular Data format.
A policy is recommended to a person when they land on an insurance website, and if the
    person chooses to fill up a form to apply, it is considered a Positive outcome (
    Classified as lead). All other conditions are considered Zero outcomes.
```

Figure 18: Dataset description for Kaggle_health-insurance-lead-prediction-raw-data.

```
Group A streptococcus (GAS) infection is a major cause of pediatric pharyngitis, and
    infection with this organism requires appropriate antimicrobial therapy.

There is controversy as to whether physicians can rely on signs and symptoms to select
    pediatric patients with pharyngitis who should undergo rapid antigen detection
    testing (RADT) for GAS .

Our objective was to evaluate the validity of signs and symptoms in the selective testing
     of children with pharyngitis.

Now, let's use machine learning to analyze whether a diagnosis can be made from the child
    's symptoms and signs.
Can we predict RADT positive?
```

Figure 19: Dataset description for Kaggle_pharyngitis.

```
Dataset Description
In this competition your task is to predict whether a passenger was transported to an
    alternate dimension during the Spaceship Titanic's collision with the spacetime
    anomaly. To help you make these predictions, you're given a set of personal records
    recovered from the ship's damaged computer system.

File and Data Field Descriptions
train.csv - Personal records for about two-thirds (~8700) of the passengers, to be used
    as training data.
PassengerId - A unique Id for each passenger. Each Id takes the form gggg_pp where gggg
    indicates a group the passenger is travelling with and pp is their number within the
     group. People in a group are often family members, but not always.
HomePlanet - The planet the passenger departed from, typically their planet of permanent
    residence.
CryoSleep - Indicates whether the passenger elected to be put into suspended animation
    for the duration of the voyage. Passengers in cryosleep are confined to their cabins
    .
Cabin - The cabin number where the passenger is staying. Takes the form deck/num/side,
    where side can be either P for Port or S for Starboard.
Destination - The planet the passenger will be debarking to.
Age - The age of the passenger.
VIP - Whether the passenger has paid for special VIP service during the voyage.
RoomService, FoodCourt, ShoppingMall, Spa, VRDeck - Amount the passenger has billed at
    each of the Spaceship Titanic's many luxury amenities.
Name - The first and last names of the passenger.
Transported - Whether the passenger was transported to another dimension. This is the
    target, the column you are trying to predict.
```

Figure 20: Dataset description for kaggle_spaceship-titanic.

