# LLMs for Semi-Automated Data Science: Introducing CAAFE for Context-Aware Automated Feature Engineering

## Abstract

As the field of automated machine learning (AutoML) advances, it becomes increasingly important to incorporate domain knowledge into these systems. We present an approach for doing so by harnessing the power of large language models (LLMs). Specifically, we introduce Context-Aware Automated Feature Engineering (CAAFE), a feature engineering method for tabular datasets that utilizes an LLM to iteratively generate additional semantically meaningful features for tabular datasets based on the description of the dataset. The method produces both Python code for creating new features and explanations for the utility of the generated features.

Despite being methodologically simple, CAAFE improves performance on 11 out of 14 datasets - boosting mean ROC AUC performance from 0.798 to 0.822 across all dataset - similar to the improvement achieved by using a random forest instead of logistic regression on our datasets.

Furthermore, CAAFE is interpretable by providing a textual explanation for each generated feature. CAAFE paves the way for more extensive (semi-)automation in data science tasks and emphasizes the significance of context-aware solutions that can extend the scope of AutoML systems. For reproducibility, we release our code and a simple demo.

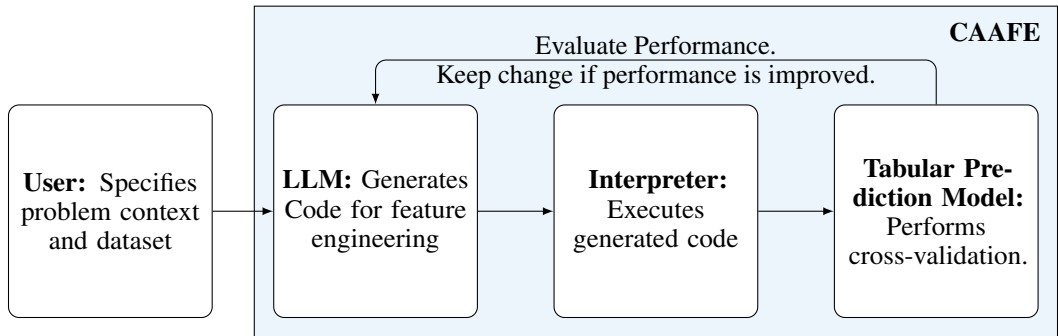

Figure 1: CAAFE accepts a dataset as well as user-specified context information and operates by iteratively proposing and evaluating feature engineering operations.

# 1 Introduction

Automated machine learning (AutoML; e.g., Hutter et al. (2019)) is very effective at optimizing the machine learning (ML) part of the data science workflow, but existing systems leave tasks such as data engineering and integration of domain knowledge largely to human practitioners. However, model selection, training, and scoring only account for a small percentage of the time spent by data scientists (roughly 23% according to the "State of Data Science"(Anaconda, 2020)). Thus, the most time-consuming tasks, namely data engineering and data cleaning, are only supported to a very limited degree by AutoML tools, if at all.

While the traditional AutoML approach has been fruitful and appropriate given the technical capabilities of ML tools at the time, large language models (LLMs) may extend the reach of AutoML to cover more of data science and allow it to evolve towards *automated data science* (De Bie et al., 2022). LLMs encapsulate extensive domain knowledge that can be used to automate various data science tasks, including those that require contextual information. They are, however, not interpretable, or verifiable, and behave less consistently than classical ML algorithms. E.g., even the best LLMs still fail to count or perform simple calculations that are easily solved by classical methods (Hendrycks et al., 2021; OpenAI Community, 2021).

In this work, we propose an approach that combines the scalability and robustness of classical ML classifiers (e.g. random forests (Breiman, 2001)) with the vast domain knowledge embedded in LLMs, as visualized in Figure 2. We bridge the gap between LLMs and classical algorithms by using code as an interface that allows LLMs to interact with classical algorithms and provides an interpretable interface to users. Our proposed method, CAAFE, generates Python code that creates semantically meaningful features that improve the performance of downstream prediction tasks in an iterative fashion and with algorithmic feedback as shown in Figure 1. Furthermore, it provides explanations for the utility of generated features. This allows for human-in-the-loop, interpretable AutoML (Lee & Macke, 2020),

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

| | | | AutoFE-Base | | CAAFE | | AutoFE-Base + CAAFE | |
| | | | AutoFeat | DFS | GPT-3.5 | GPT-4 | GPT-4 + AF | GPT-4 + DFS |
|---|---|---|---|---|---|---|---|---|
| Log. Reg. | Mean | 0.749 ±.14 | 0.754 ±.14 | 0.764 ±.14 | 0.75 ±.14 | 0.769 ±.13 | 0.781 ±.14 | **0.784** ±.14 |
| | Mean Rank | 5.11 | 4.21 | 3.36 | 4.79 | 4.31 | 3.52 | **2.69** |
| Rand. Forest | Mean | 0.783 ±.13 | 0.782 ±.12 | 0.781 ±.14 | 0.783 ±.13 | 0.801 ±.14 | **0.808** ±.14 | 0.808 ±.14 |
| | Mean Rank | 4.44 | 4.41 | 4.14 | 4.38 | 4.28 | **3.15** | 3.21 |
| TabPFN | Mean | 0.798 ±.14 | 0.797 ±.14 | 0.791 ±.15 | 0.799 ±.14 | **0.822** ±.14 | 0.821 ±.14 | 0.818 ±.14 |
| | Mean Rank | 4.39 | 4.4 | 4.76 | 4.26 | **3.06** | 3.27 | 3.86 |

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

**Incorporating Classical AutoFE Methods** Classical AutoFE methods can readily be combined with our method, one simply runs CAAFE first and then lets a classical AutoFE method find further feature extensions, as we did in Table 5. For less powerful downstream classifiers, namely Logistic Regression and Random Forests, we observe that applying AutoFE additionally to CAAFE improves performance further. The AutoML method TabPFN on the other hand cannot be improved by applying classical AutoFE. This discrepancy might stem from the larger hypothesis space (complexity) of TabPFN, it can get all necessary information from the data directly.For all combinations of classifiers and additional AutoFE methods, we can see that CAAFE improves performance on average.

**Feature Engineering Strategies** Table 3 shows a diverse set of examples of feature engineering strategies applied by our method. We show examples where

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

| ⟨$Kaggle$⟩ health-insurance | 0.5745 ±.02 | **0.5805** ±.03 | 0.5753 ±.02 | 0.5745 ±.02 | 0.5748 ±.02 | 0.5777 ±.03 | 0.5782 ±.03 |
| ⟨$Kaggle$⟩ pharyngitis | 0.6976 ±.03 | 0.6976 ±.03 | 0.6976 ±.03 | 0.6976 ±.03 | **0.7078** ±.04 | 0.7073 ±.04 | 0.6976 ±.03 |
| ⟨$Kaggle$⟩ kidney-stone | 0.7883 ±.04 | 0.7856 ±.04 | 0.7929 ±.04 | 0.7873 ±.04 | 0.7903 ±.04 | 0.7875 ±.04 | **0.7967** ±.03 |
| ⟨$Kaggle$⟩ spaceship-titanic | 0.838 ±.02 | 0.8486 ±.02 | 0.8443 ±.02 | 0.8383 ±.02 | 0.8405 ±.02 | **0.853** ±.02 | 0.8486 ±.02 |
| pc1 | 0.9035 ±.01 | 0.9046 ±.01 | 0.9035 ±.01 | 0.9087 ±.02 | **0.9093** ±.01 | 0.908 ±.01 | 0.9035 ±.01 |
| tic-tac-toe | 0.6989 ±.08 | 0.6989 ±.08 | 0.6291 ±.10 | 0.6989 ±.08 | **0.9536** ±.06 | **0.9536** ±.06 | 0.938 ±.06 |
| Mean ROC | 0.798 ±.05 | 0.7966 ±.05 | 0.7913 ±.05 | 0.7987 ±.05 | **0.8215** ±.04 | 0.8209 ±.05 | 0.8176 ±.04 |
| Mean Rank | 4.68 | 4.75 | 5.29 | 4.36 | **2.39** | 2.57 | 3.96 |