# OpenReview forum: "Large Language Models for Automated Data Science: Introducing CAAFE for Context-Aware Automated Feature Engineering"
_NeurIPS.cc/2023/Conference — NeurIPS 2023 poster_

### Official Review · Reviewer_uCcq · 2023-06-27

**Soundness:** 3 good
**Presentation:** 3 good
**Contribution:** 2 fair
**Rating:** 6
**Confidence:** 4

**Summary:**

The paper proposes CAAFE, a Context aware automated feature engineering approach to support auto ML by utilizing Large Language Models (LLMs). Proposed method generates new features or transforming the feature space from tabular data by using LLMs and incorporate them to train new models. Transforming the feature space can be an iterative process where in each step LLM generates code that is run on training and evaluation sets, followed by fitting a new model and evaluating its performance. LLM receives a set of instructions through a prompt which includes semantic information about the data set, feature names, their data types, useful stats about features, and a few sample values.
The paper reports experimental results on two sets of data sets: 1- OpenML data sets published before Sept 2021 (GPT 3.5 and 4 training date); 2- Kaggle data sets published after Sept 2021. Results show that CAAFE can improve the performance of a state of the art classifier. They also show that CAAFE can help build models that outperform models built with the help of other state of the art automatic feature generation.

**Strengths:**

Although the seminal idea proposed in the paper is rather straightforward, it's novelty makes it a worthwhile body of research for consideration.

- originality: The work is novel and original. It introduces the use of LLMs for auto ML through adding contextual information which is less explored.
- quality: The paper is technically sound and experimental results support the claims. Limitation and risks of the work are also clearly called out.
- clarity: The paper is easy to read and follow and the contributions are discussed clearly.
- significance: Experimental results show LLM's usefulness for feature engineering aspect of AutoML which testifies the significance of the work. For more detailed comments on submission's significance refer to the limitation section.

**Weaknesses:**

- Alternative feature generation methods that are considered in the experiments are limited. AutoFE is the main alternative method. Other techniques such as PCA or deep sparse feature learning (authors have mentioned that with small data sets they don't expect much benefit from deep learning methods but it would be interesting to emperically validate this hypothesis for the studied data sets) is not considered.
- The human-in-the-loop aspect of CAAFE is not fully discussed. For example, one of the implications of having human in the loop is correcting the potential mistakes that LLM may make. In the paper it is not discussed how many times the generated code by LLM needed to be revised by human, not necessarily due to run time errors but because of the semantic issues in the generated features.

**Questions:**

- How do you make sure the generated feature makes sense? e.g. https://github.com/cafeautomatedfeatures/CAFE/blob/main/data/generated_code/airlines_v3_0_code.txt why Euclidean distance between airports even make sense? Based on the data set description it is not clear if from or to airport attributes represent distance to a canonical point.
- Minor, line 195, please fix the question marks in "Table ??".

**Limitations:**

The main limitation of the work is stated by authors as "We focus on small datasets with up to 2,000 samples in total, because feature engineering is most important and significant for smaller datasets". As stated, the proposed approach is mainly beneficial for smaller data sets which makes the method less useful for a lot of real world and specially industrial applications.
Along the same lines, as clearly mentioned by author(s), another limit of the work is the possibility of exceeding the prompt maximum length.

---

> ### Author Rebuttal · Authors · 2023-08-09
>
> We deeply appreciate the time and effort taken by you to evaluate our paper. We've carefully considered each point raised and aim to address them comprehensively below.
>
> For this year's NeurIPS revisions, it is not possible to upload a modified paper, but only a one-page rebuttal PDF. Hence, we've outlined the changes we'll implement in the final paper within our responses.
>
> [Weaknesses]
>
> *[1] Alternative feature generation methods that are considered in the experiments are limited. AutoFE is the main alternative method. Other techniques such as PCA or deep sparse feature learning (authors have mentioned that with small data sets they don't expect much benefit from deep learning methods but it would be interesting to emperically validate this hypothesis for the studied data sets) is not considered.*
>
> We evaluated more baselines: (1) AutoFE methods: FETCH and OpenFE (2) AutoML methods: Autosklearn and Autogluon (Which perform feature engineering as part of their pipeline. Here, the effect of additional AutoFE should be smaller and could be negative). Autosklearn e.g. also includes PCA among others for feature engineering. Please see the first point in our general rebuttal, as well as Figure 2 in the rebuttal PDF for detailed results You can find detailed results in Table 2 (one-page rebuttal PDF) in the PDF. TLDR: CAAFE (GPT-4) + TabPFN is strongest among all methods and adds performance to AutoML methods. Using TabPFN as a classifier, a critical difference diagram shows statistical significance of CAAFE to all baselines. For FETCH, we could only evaluate it optimized for Logistic Regression due to the large computational cost (up to 24h / dataset / seed)..
>
> *[2] The human-in-the-loop aspect of CAAFE is not fully discussed. For example, one of the implications of having human in the loop is correcting the potential mistakes that LLM may make. In the paper it is not discussed how many times the generated code by LLM needed to be revised by human, not necessarily due to run time errors but because of the semantic issues in the generated features.*
>
> We mention that human-in-the-loop AutoML could be possible since semantic explanations can be more interpretable. An empirical evaluation of a human-in-the-loop concept would require running experiments with a cohort of users interacting with our algorithm. With our resources this is not feasible in terms of cost and time. This could be a followup to our work, likely coming from the industry, if commercialization is a goal.
>
> [Questions]
>
> *[1] How do you make sure the generated feature makes sense? e.g. https://github.com/cafeautomatedfeatures/CAFE/blob/main/data/generated_code/airlines_v3_0_code.txt why Euclidean distance between airports even make sense? Based on the data set description it is not clear if from or to airport attributes represent distance to a canonical point.*
>
> This file has been generated by GPT-3.5 which produces less meaningful, coherent and empirically useful features (All files with "v3" in the name are generated by GPT-3.5, "v4" is GPT-4 - which we did not mention!). Evaluating if generated features are semantically meaningful is hard, since such an evaluation itself would be subjective. Once again, it would require a cohort of raters, judging subjectively. We do report examples in our paper and in the supplements and address in our limitations that there is no guarantee for features to be meaningful - however, looking at a few examples you quickly see that many are - while not perfect this is a significant leap forward to classical feature engineering.
>
> We added the following to our limitations:
> "LLMs, at times, exhibit a phenomenon known as "hallucinations.", where models produce inaccurate or invented information. Within CAAFE, this might result in the generation of features and associated explanations that appear significant and are logically presented, even though they may not be grounded in reality. Such behavior can be problematic, especially when individuals place trust in these systems for essential decision-making or research tasks."
>
> *[2] Minor, line 195, please fix the question marks in "Table ??".*
>
> Done, thank you! :)

---

> > ### Comment · Reviewer_uCcq · 2023-08-19
> >
> > I'd like to thank the authors for their responses to my questions/comments. Great to see the experimental results from considering more baselines.

---

### Official Review · Reviewer_Cmyk · 2023-07-01

**Soundness:** 4 excellent
**Presentation:** 4 excellent
**Contribution:** 2 fair
**Rating:** 6
**Confidence:** 4

**Summary:**

This paper presents Context-Aware Automated Feature Engineering (CAAFE) for integrating domain knowledge into the AutoML process using LLMs. CAAFE automates feature engineering for tabular datasets, generating Python code that generates semantically meaningful features based on the dataset and a textual description of the data.  The study demonstrates the effectiveness of CAAFE on a range of benchmark tasks.



**Strengths:**

- Well written and easy to follow. Thank you!
- Simple and very applicable to real-world scenarios!

**Weaknesses:**

- If set up correctly, the method really can't fail. Features are only added, if they improve the objective. Thus the scientific value is somewhat limited to showing that LLMs can create features that improve performance (expected) and the extent to which (here is where I see most value). Therefore comparison to other feature engineering methods seems really important and could be more extensive.
- Fig 2 is not clear. Is this a flow from left to right? How does the bottom part relate to the boxes? Please explain better.

Suggestions:
- I don't think the phrase "code as interface" is self-explanatory. Maybe can be introduced explicitly
- In section 2.1., the sentence "GPT-4 is a deep neural network that uses a transformer architecture" - may not actually be correct. It is rumoured that GPT4 consists of multiple models and uses some mixture of experts methodology. In any case, this information is not published which might be worthwhile pointing out.
- l195, table reference is missing

**Questions:**

- What about the dataset where performance does not improve or even weaken? Is there an explanation.

**Limitations:**

- The main limitation is more extensive study of competing approaches. In essence the authors work really doesn't answer any open scientific questions.

---

> ### Author Rebuttal · Authors · 2023-08-09
>
> We deeply appreciate the time and effort taken by you to evaluate our paper. We've carefully considered each point raised and aim to address them comprehensively below. Thank you for appreciating the paper presentation!
>
> We would like to address your questions and our changes in response to them in detail below. We would also appreciate you reading through the general reviewer feedback, which outlines the changes we made and might address open questions that go beyond what you have asked for.
>
> For this years NeurIPS revisions, it is not possible to upload a modified paper, but only upload a one page rebuttal with additional figures. When changes in text of the main paper were made, we replied these changes to each reviewer individually.
>
> We were a bit surprised at the low contribution score - we believe this paradigm of introducing context / semantic into automated machine learning can be quite meaningful and it is highly original (Reviewer uCcq: "​​Although the seminal idea proposed in the paper is rather straightforward, it's novelty makes it a worthwhile body of research for consideration.", Reviewer Y55o: "If I'm not mistaken, the first positive showing of LLMs in tabular data", viewer ndjC: "To the best of my knowledge, this is the first work to employ LLMs (GPT) for automating feature engineering.")
>
> [Weaknesses]
>
> *[1] "If set up correctly, the method really can't fail. Features are only added, if they improve the objective. Thus the scientific value is somewhat limited to showing that LLMs can create features that improve performance (expected) and the extent to which (here is where I see most value)."*
>
> A featurization is only validated on the training data, but performance is measured on separate test data. Overfitting can occur in the same way as it does in training a model - a featurization can be viewed as just another step in the model construction. When the number of provided samples is small or the number of tested featurizations is large, the risk of overfitting becomes larger. By considering semantically meaningful features CAAFE contains a prior for features that are more likely to generalize to the test set. Also, it reduces computational complexity by considering useful features more quickly. Looking at the generated code, these featurizations can contain up to 9 features at the same time - imagine the likelihood of such a feature when randomly permuting.
>
> *[2] "Therefore comparison to other feature engineering methods seems really important and could be more extensive."*
>
> We evaluated more baselines: (1) AutoFE methods: FETCH and OpenFE (2) AutoML methods: Autosklearn and Autogluon (Which perform feature engineering as part of their pipeline. Here, the effect of additional AutoFE should be smaller and could be negative). You can find detailed results in Table 2 (one-page rebuttal PDF) in the PDF. TLDR: CAAFE (GPT-4) + TabPFN is strongest among all methods and adds performance to AutoML methods. Using TabPFN as a classifier, a critical difference diagram shows statistical significance of CAAFE to all baselines. For FETCH, we could only evaluate it optimized for Logistic Regression due to the large computational cost (up to 24h / dataset / seed).
>
> *[3] Fig 2 is not clear. Is this a flow from left to right? How does the bottom part relate to the boxes? Please explain better.*
>
> Thank you for the feedback! We removed the arrows and adapted the caption: "Figure 2: Data Science pipeline, inspired by De Bie et al. (2022). CAAFE allows for automation of semantic data engineering, while LLMs could provide even further automation: (1) Context specification is user driven (2) exploitation and data engineering can be automated through LLMs (3) model building can be automated by classical AutoML approaches". We hope this would make this diagram clearer.
>
> [Suggestions]
>
> *[1] I don't think the phrase "code as interface" is self-explanatory. Maybe can be introduced explicitly*
>
> We did so in our updated manuscript, thank you! 🙂
> "We bridge the gap between LLMs and classical algorithms by using code as an interface between them: LLMs generate code that modifies input datasets, these modified datasets can then be processed by classical algorithms."
>
> *[2] In section 2.1., the sentence "GPT-4 is a deep neural network that uses a transformer architecture" - may not actually be correct. It is rumoured that GPT4 consists of multiple models and uses some mixture of experts methodology. In any case, this information is not published which might be worthwhile pointing out.*
>
> We did so in our updated manuscript, we write: "The architecture of GPT-4 is not published, it is likely based on a deep neural network that uses a transformer architecture [...]"
>
> *[3] l195, table reference is missing*
>
> Thank you, done
>
> [Questions]
>
> *[1] What about the dataset where performance does not improve or even weaken? Is there an explanation.*
>
> We believe an answer to this is contained in our reply to your first point of weaknesses.
> [Limitations]
>
> *[2] The main limitation is more extensive study of competing approaches.*
>
> We believe an answer to this is contained in our reply to your second point of weaknesses.

---

> > ### Comment · Reviewer_Cmyk · 2023-08-14
> > **Thank you...**
> >
> > Thank you for addressing reviewer comments in depths. Given the additional evaluation, I will adjust my overall rating to 7.

---

> > > ### Author Response · Authors · 2023-08-16
> > > **Thank you for your feedback**
> > >
> > > Thank you for your feedback and intention to update your score. We truly appreciate the time and effort spent in reviewing our submission. You can update the score by clicking on "Edit Review" on your original review - we would very much appreciate that.

---

### Official Review · Reviewer_ndjC · 2023-07-05

**Soundness:** 2 fair
**Presentation:** 4 excellent
**Contribution:** 3 good
**Rating:** 3
**Confidence:** 4

**Summary:**

In this paper, the authors propose a feature engineering method that builds upon LLMs. Features are engineering in an iterative process of prompting the LLM to generate code for new features, evaluating the features with a ML model and generating a new prompt. Hence, the overall approach follows the common wrapper approach. In their empirical evaluation they find feature engineering via LLMs to outperform no feature engineering.

**Strengths:**

- To the best of my knowledge, this is the first work to employ LLMs (GPT) for automating feature engineering.
- The general idea seems to work out quite nicely and compare comeptitively to other feature engineering methods and preferably over engineering no features.
- The paper is well written and easy to follow. The overall presentation is really excellent - maybe due to the use of ChatGPT for the editing?

**Weaknesses:**

Although the idea in general is quite interesting, I have several doubts regarding this work. The first and foremost doubt is about how to sort this work into the already existing literature and whether it really establishes a new state of the art. In particular, I am thinking about the following two works which have been missed also during the discussion of related works:
Li, Liyao, et al. "Learning a Data-Driven Policy Network for Pre-Training Automated Feature Engineering." The Eleventh International Conference on Learning Representations. 2022.
Zhang, Tianping, et al. "OpenFE: Automated Feature Generation beyond Expert-level Performance." arXiv preprint arXiv:2211.12507 (2022).
(https://openreview.net/forum?id=1H1irbEaGV)
This might be due to these works being rather recent. However, preprints of these papers have already been available for some time. I believe a comparison to these methods is inevitable to prove the real benefit of LLMs and whether the semantic information of feature names is really needed to come up with better features.

Another issue with the paper is that it claims that LLMs are taking advantage of the semantical information of a dataset, e.g., the names of the features etc. But this is not substantiated in the experiments nor visible in the explanations generated in the examples.

Moreover, I am a little bit doubtful to what extent the explanations of an LLM can be meaningful when LLMs base the text generated on the likelihood of the next word. In particular, logical argumentation does not seem to be a strengths of GPT so far and the flaws in logical reasoning can be made easily visible. E.g. testing with a sentence like "The [doctor/nurse] yelled at the [doctor/nurse] because she was late. Who was late?" and asking for an explanation for the decision. Why should the explanations be more meaningful for datasets?

Speaking about the semantics that are supposedly extracted from feature names, it would have been interesting to see whether this information is indeed used, e.g., by blinding feature names. Neither in the main paper nor in the supplement I could find experiments that aim at answering these questions. However, these experiments lacking mean an unproved claim made in the paper.

Minor:
- Reference to table in line 195 broken.

**Questions:**

1. How does CAAFE compare to the SOTA in automated feature engineering?
2. What makes CAAFE semi-automated? As far as I understood, CAAFE only requires an input by the user once in the beginning to give a description of the dataset. If this is the reason for semi-automation are not every AutoML tools out there semi-automated?
3. Do LLMs in CAAFE really leverage semantic information about the dataset?
4. As the authors already state in their Broader Impact Statement there is a certain risk that biases contained in LLMs transfer to the engineered features and that CAAFE then builds features based on these biases. Should not every feature engineering approach based on LLMs directly incorporate mechanisms to prevent to leverage such biases? Especially since such biases might be very subtle.

**Limitations:**

The authors prove a broader impact statement estimating what effect and potential negative societal impact there might be.

---

> ### Author Rebuttal · Authors · 2023-08-09
>
> We deeply appreciate the time and effort taken by you to evaluate our paper. We've carefully considered each point raised and aim to address them comprehensively below. Thank you for appreciating the paper presentation - to answer your question about GPT-4 use: We did indeed use GPT-4 to prepare the work, which has been especially useful in creating and refining Latex plots! We mention this in the Acknowledgements section.
>
> For this year's NeurIPS revisions, it is not possible to upload a modified paper, but only a one-page rebuttal PDF. Hence, we've outlined the changes we'll implement in the final paper within our responses.
>
> ### Weaknesses
>
> 1. *You ask for further comparisons to FETCH and OpenFE.*
>
> We evaluated more baselines: (1) We additionally compared to the AutoFE methods FETCH and OpenFE, and (2) we additionally used the AutoML methods Autosklearn and Autogluon as base classifiers, which perform feature engineering as part of their pipeline. You can find detailed results in Table 2 (one-page rebuttal PDF) in the PDF. TLDR: We find that one of the two CAAFE variants is the best feature engineering method for all base classifiers in both mean and mean rank AUROC, but for logistic regression. Additionally, we find that CAAFE (GPT-4) + TabPFN is the strongest among all methods. Using TabPFN as a classifier, a critical difference diagram shows a statistically significant edge of CAAFE over all baselines. For FETCH, we could only evaluate it optimized for Logistic Regression due to the large computational cost (up to 24h / dataset / seed).
>
> 2. *"Another issue with the paper is that it claims that LLMs are taking advantage of the semantic information of a dataset, e.g., the names of the features etc. But this is not substantiated in the experiments"*
>
> We performed an additional ablation study to test this hypothesis. We blind feature names and dataset descriptions, i.e. CAAFE is applied to the datasets without contextual information. We find a strong drop in performance from an average AUROC of 0.822 to 0.8 over all datasets for GPT-4. Please, see Table 1 in the rebuttal PDF for detailed results.
>
> Looking at the generated features very strongly highlights the usefulness of semantic information. Consider e.g. the tic-tac-toe dataset: The generated features are based on up to 9 features - a useful combination of that many features is very hard to find with random search but given contextual information this feature is quickly discovered.
>
> 3. *"Moreover, I am a little bit doubtful to what extent the explanations of an LLM can be meaningful when LLMs base the text generated on the likelihood of the next word. [..]"*
>
> Generated explanations do not need to be complex, in many cases this can be a simple explanation of what the generated code does. This can already be quite useful to non-technical users.
>
> We published all generated features for you to inspect at https://github.com/cafeautomatedfeatures/CAFE/tree/main/data/generated_code. They show that generated feature explanations are often of high quality and include semantic information. This quality of the provided explanations often surprised us as well. We have in the meantime received a significant amount of feedback that highlights that this interpretability is going to be especially valuable to practitioners. Take the following generation for balance_scale as an example: https://github.com/cafeautomatedfeatures/CAFE/blob/main/data/generated_code/balance-scale_v4_0_code.txt
> Here the task is predicting if a scale tips.
>
> We do, however, want to address the issue of false explanations and to our limitations:
> LLMs, at times, exhibit a phenomenon known as "hallucinations", where models produce inaccurate or invented information. Within CAAFE, this might result in the generation of features and associated explanations that appear significant and are logically presented, even though they may not be grounded in reality. Such behavior can be problematic, especially when individuals place trust in these systems for essential decision-making or research tasks.
>
> ### Questions
>
> 1. *How does CAAFE compare to the SOTA in automated feature engineering?*
>
> We believe an answer is contained in our reply to your first point of weaknesses.
>
> 2. *What makes CAAFE semi-automated? As far as I understood, CAAFE only requires an input by the user once in the beginning to give a description of the dataset. If this is the reason for semi-automation are not every AutoML tools out there semi-automated?$
>
> Our intention was the following: Classical AutoML deals with model selection and technical optimizations which do not require semantic information. The term AutoDS or Automated Data Science often refers to a more extensive automation of the Data Science stack, where AutoML just covers a portion. We chose the term semi-automated Data Science since we are not automating all of the data science stack, but another part of it, while highlighting that this is a step towards AutoDS, i.e. full automation of the DS pipeline. The naming might be confusing, however, and we are considering a slight change.
>
> 3. *Do LLMs in CAAFE really leverage semantic information about the dataset?*
>
> See the reply to Weakness 2.
>
> 4. *As the authors already state in their Broader Impact Statement there is a certain risk that biases contained in LLMs transfer to the engineered features and that CAAFE then builds features based on these biases. Should not every feature engineering approach based on LLMs directly incorporate mechanisms to prevent to leverage such biases? Especially since such biases might be very subtle.*
>
> We have formulated a detailed answer in our reply to the ethics reviewer Ce9G and refer you there for an in-depth discussion of biases as we only have limited space in each rebuttal post.

---

> > ### Comment · Reviewer_ndjC · 2023-08-21
> > **Re: Rebuttal by Authors**
> >
> > The elaborate rebuttal is very much appreciated and my concerns regarding CAAFE were lowered by the authors' response and additional experiments. However, I, unfortunately, cannot see the ethics reviewer's comments and thus cannot judge to what extent these issues have been resolved.
> >
> > Yet, it is relatively hard to judge based on the aggregated means without any standard deviation whether the drop in performance is indeed substantial and outside the regime of the standard error.
> > Furthermore, for GPT-3.5 it seems as if it cannot really leverage the semantic information of the feature descriptions at all? How can this be explained?

---

> > > ### Author Response · Authors · 2023-08-22
> > >
> > > Thank you very much for your reply!
> > >
> > > Unfortunately the ethics reviewer has not replied so far - we have provided a two part answer with examples and an improvement Broader impact section.
> > >
> > > *"Yet, it is relatively hard to judge based on the aggregated means without any standard deviation whether the drop in performance is indeed substantial and outside the regime of the standard error."*
> > >
> > > In our rebuttal PDF, we include Critical Difference Diagrams in Figure 3, which implement the Wilcoxon test with a Bonferroni multiple testing correction. This test corrects for multiple testing between many all our baselines and makes this part of the testing procedure.
> > >
> > > Critical difference (CD) diagrams are a powerful tool to compare outcomes of multiple treatments over multiple observations. For instance, in machine learning research we often compare the performance of multiple methods over multiple data sets (i.e., observations).
> > >
> > > A diagram like the one above concisely represents multiple hypothesis tests that are conducted over the observed outcomes. Before anything is plotted at all, the **Friedman test** tells us whether there are significant differences at all. If this test fails, we have not sufficient data to tell any of the treatments apart and we must abort. If, however, the test sucessfully rejects this possibility we can proceed with the post-hoc analysis. In this second step, a **Wilcoxon signed-rank test** tells us whether each pair of treatments exhibits a significant difference.
> > >
> > > Since we are testing multiple hypotheses, we must adjust the Wilcoxon test with Bonferroni's method. For each group of treatments which we can not distinguish from the **Bonferroni-adjusted Wilcoxon test**, we add a thick line to the diagram.
> > >
> > > *Furthermore, for GPT-3.5 it seems as if it cannot really leverage the semantic information of the feature descriptions at all? How can this be explained?*
> > >
> > > This highlights how much better GPT-4 is able to integrate semantic information. We see that GPT-3.5 often uses semantic information is a false way and hallucinates. See this example on the airplanes dataset, where it uses Airport IDs to measure distance - here the idea seems suitable to calculate distances but the details are missed: this information is not contained within the IDs.
> > >
> > > ```
> > > # Usefulness: Distance between airports can be a useful feature for predicting flight delays. Longer distances may have more potential for delays due to weather, air traffic control, etc.
> > > # Input samples: 'AirportFrom': [225.0, 39.0, 5.0], 'AirportTo': [11.0, 7.0, 60.0]
> > > df['Distance'] = ((df['AirportFrom'] - df['AirportTo'])**2)**0.5
> > > ```

---

> > > > ### Comment · Reviewer_ndjC · 2023-08-22
> > > > **Re: Officiel Comment by Authors**
> > > >
> > > > I know these significance tests and critical distance diagrams but you do not provide significance tests for the experiments regarding semantical blinding. Hence, my question is still open to what extent the use of semantics from the dataset description is significantly better than the ones with semantic blinding.

---

### Official Review · Reviewer_Y55o · 2023-07-06

**Soundness:** 2 fair
**Presentation:** 3 good
**Contribution:** 3 good
**Rating:** 6
**Confidence:** 5

**Summary:**

This paper presents a novel approach to automated feature engineering utilizing large language models (LLMs). Authors propose to use LLMs to generate code for feature generation. In the proposed method CAAFE, LLM is given a prompt with dataset description and a task of writing code for feature generation, the new feature is accepted if the validation score improves after training an ML model on a dataset containing the new feature. This procedure is repeated.

The paper demonstrates the effectiveness of the approach by testing on 10 public datasets from OpenML and 4 datasets from Kaggle (less likely to leak into the LLM training set) and slightly improving the overall classification accuracy of the strong tabular classification model TabPFN

**Strengths:**

- I find the idea of incorporating LLMs via code generation to tabular data automatic feature engineering clever and clean
- Having interpretable features is a big plus and an advantage over prior automatic methods
- If I'm not mistaken, the first positive showing of LLMs in tabular data (where using an LLM seem to improve the performance of an already strong tabular model)
- Percussion taken to create a non-leaked test-set

**Weaknesses:**

The major issue (that's why the score is not as high as it could have been for me) is the lacking evaluation. Both in terms of baselines/complementary methods and datasets.
- Authors compare and complement the proposed method with AutoFeat `[1]` and DFS `[2]`, while much more effective automatic feature engineering methods like FETCH `[3]` and OpenFE `[4]` exist.
- Also, in the above-mentioned papers, the evaluations are much more extensive, including more datasets (also regression datasets) and base models. Implementing CAAFE in those benchmarks would be a very big plus for the validation of the proposed method.



**Questions:**

- What do the bold entries mean in the table? The standard deviation values are pretty large, but results are bolded. Were stat tests used?
- Does performance saturate after 10 iterations? What is the reason for stopping there?
- Do you update the prompt with the generated feature descriptions for new iterations (I think info is missing in the paper, but this seems important)?
- Could you report some numerical quantification of features generated by CAAFE, like feature importance ranking?
- Minor issues:
  - line 159: should probably cite TabPFN
  - line 195: broken reference

**Limitations:**

- I think the fact that method requires dataset and feature descriptions is a limitation
- Looking at the generated code, some features and their usefulness explanations are, almost surely, hallucinations. I think It's important to mention hallucinations in the limitations section to represent this.

**References**
- `[1]` Franziska Horn, Robert Pack, and Michael Rieger. The autofeat python library for automatic feature engineering and selection.
- `[2]` James Max Kanter and Kalyan Veeramachaneni. Deep feature synthesis: Towards automating data
science endeavors
- `[3]` Li, Liyao, et al. "Learning a Data-Driven Policy Network for Pre-Training Automated Feature Engineering."
- `[4]` Zhang, Tianping, et al. "OpenFE: Automated Feature Generation beyond Expert-level Performance."

---

> ### Author Rebuttal · Authors · 2023-08-09
>
> We deeply appreciate the time and effort taken by you to evaluate our paper. We've carefully considered each point raised and aim to address them comprehensively below.  We are especially happy that you appreciate the interpretability of our generated features. We have in the meantime received lots of feedback, that this point is going to be especially valuable to practitioners.
>
> We would also appreciate you reading through the general reviewer feedback, which outlines the changes we made and might address open questions that go beyond what you have asked for.
>
> For this year's NeurIPS revisions, it is not possible to upload a modified paper, but only a one-page rebuttal PDF. Hence, we've outlined the changes we'll implement in the final paper within our responses.
>
>
> [Weaknesses]
>
> *[1] "Authors compare and complement the proposed method with AutoFeat [1] and DFS [2], while much more effective automatic feature engineering methods like FETCH [3] and OpenFE [4] exist."*
>
> We evaluated more baselines: (1) AutoFE methods: FETCH and OpenFE (2) AutoML methods: Autosklearn and Autogluon (Which perform feature engineering as part of their pipeline. Here, the effect of additional AutoFE should be smaller and could be negative). You can find detailed results in Table 2 (one-page rebuttal PDF) in the PDF. TLDR: CAAFE (GPT-4) + TabPFN is strongest among all methods and adds performance to AutoML methods. Using TabPFN as a classifier, a critical difference diagram shows statistical significance of CAAFE to all baselines. For FETCH, we could only evaluate it optimized for Logistic Regression due to the large computational cost (up to 24h / dataset / seed).
>
> *[2] "Also, in the above-mentioned papers, the evaluations are much more extensive, including more datasets (also regression datasets) and base models. Implementing CAAFE in those benchmarks would be a very big plus for the validation of the proposed method."*
>
> For CAAFE we have a special set of requirements which makes its evaluation more challenging than previous approaches: (1) CAAFE requires meaningful dataset descriptions, which are not available for all of these previous datasets (2) as stated in our limitations, the cost of applying CAAFE rises linearly with the number of features in a dataset, thus we focus on datasets with less than 20 features.
> We believe that contextual prediction problems, where a dataset is modelled with a description of its use case, will grow in significance as deep learning becomes more multimodal and language models are applied to more modalities. A larger benchmark of datasets with interesting and varied contextual information will be vital to evaluate these works, especially when comparing multiple contextual algorithms.
>
> [Questions]
>
> *[1] "What do the bold entries mean in the table? The standard deviation values are pretty large, but results are bolded. Were stat tests used?"*
>
> We simply bolded entries with the largest mean values. Standard deviations are across data-splits and can thus be quite large. We added an explanation to the table describing which entries are lighted. Also, we use a statistical significance test, comparing AutoFE using a critical difference diagram, that employs a Wilcoxon test for statistical significance (taking into account multiple testing for multiple tested methods). See our answer to Weakness [1].
>
> *[2] "Does performance saturate after 10 iterations? What is the reason for stopping there?"*
>
> Supplement F shows the cost and performance improvement when running CAAFE for 1-10 iterations. We see that performance is rising even after 10 iterations. 10 was simply the parameter that we started with. We did, however, not evaluate more iterations since we ran out of budget and could not repeat all experiments with another iteration parameter.
>
> *[3] "Do you update the prompt with the generated feature descriptions for new iterations (I think info is missing in the paper, but this seems important)?"*
>
> The prompt in each iteration contains the previously generated code and the performance after executing that feature engineering step. Thus, CAAFE can learn from previous code operations. This important information has, indeed been missing from our work, and we did include it in our new version. In line 141, we add:
> "F: Any code generated by CAAFE in previous iterations, as well as ROC AUC and accuracy evaluated on the validation splits."
>
> *[4] "Could you report some numerical quantification of features generated by CAAFE, like feature importance ranking?"*
>
> Yes surely - we had a plot prepared for this that shows the feature importance of multiple datasets. Unfortunately, due to the page limit (1 page) for figures in this rebuttal, we could not include these plots... We will include this plot in the final work. On average the mean importance of generated features in our trained random forests was 1.49 times that of the original features.
>
> *[5] Minor issues: line 159: should probably cite TabPFN, line 195: broken reference*
>
> Thank you, we have addressed these issues :)
>
> [Limitations]
>
> *[1] Looking at the generated code, some features and their usefulness explanations are, almost surely, hallucinations. I think It's important to mention hallucinations in the limitations section to represent this.*
>
> We added the following section to our conclusion:
> LLMs, at times, exhibit a phenomenon known as "hallucinations.", where models produce inaccurate or invented information. Within CAAFE, this might result in the generation of features and associated explanations that appear significant and are logically presented, even though they may not be grounded in reality. Such behaviour can be problematic, especially when individuals place trust in these systems for essential decision-making or research tasks.

---

### Official Review · Reviewer_QFgW · 2023-07-07

**Soundness:** 2 fair
**Presentation:** 3 good
**Contribution:** 3 good
**Rating:** 4
**Confidence:** 5

**Summary:**

The paper proposed Context-Aware Automated Feature Engineering (CAAFE) approach to integrate new features learned from dataset descriptions using large language models into AutoML process for tabular datasets. The proposed approach was evaluated using 14 datasets.

**Strengths:**

The paper is well-written with clear research motivation, experiment design and results illustration. It's an interesting attempt to try to integrate large language models into AutoML process.
Code scripts are provided.

**Weaknesses:**

1. Overall the use case of adding features from data description is trivial, even using large language models or human-in-the-loop concept. The proposed feature engineering process through prompt generation from the data description or any semantic information looks more of heuristic rules in featurization, which could be performed without large language models. LLM plays the role of automated code generation, instead of machine learning feature engineering approaches (e.g. PCA) incorporated in most AutoML systems.
2. The paper fails to catch up with the latest trends of AutoML systems. Most of existing AutoML systems will perform automated feature engineering, e.g. H2O AutoML, TPOT, AutoGluon, AutoSklearn, etc. Missing those AutoML baselines significantly eliminate the value of the proposed approach in modeling effectiveness.
3. In Table 1, the increased CAAFE performance is relatively trivial for most datasets, except the tic-tac-toe, which significantly impact the mean ROC AUC.

**Questions:**

Highly recommend the author to run the data experiments using the existing AutoML systems (listed in weakness part, they are all open source) without LLM featurizations from dataset description as baselines.

**Limitations:**

Yes. The author addressed the limitations.

---

> ### Author Rebuttal · Authors · 2023-08-09
>
> We deeply appreciate the time and effort taken by Reviewer QFgW to evaluate our paper. We've carefully considered each point raised and aim to address them comprehensively below. We would also appreciate you reading through the general reviewer feedback, which outlines the changes we made and might address open questions that go beyond what you have asked for.
> For this year's NeurIPS revisions, it is not possible to upload a modified paper, but only upload a one-page rebuttal with additional figures. So we posted changes that we will make in the final paper as part of our replies.
>
> ### Weaknesses
>
> 1. *"Overall the use case of adding features from data description is trivial, even using large language models or human-in-the-loop concept. The proposed feature engineering process through prompt generation from the data description or any semantic information looks more of heuristic rules in featurization, which could be performed without large language models. LLM plays the role of automated code generation, instead of machine learning feature engineering approaches (e.g. PCA) incorporated in most AutoML systems."*
>
> We believe that there is strong evidence that the generated features are not trivial and significantly benefit from the LLMs knowledge.
>
> Baseline approaches such as DFS, AutoFeat, FETCH and OpenFE perform worse than our approach, and we do not believe that features these approaches propose are trivial. Also, CAAFE improves even the performance of AutoML methods, such as Autogluon and Autosklearn, which already have built-in feature engineering. Please, see Table 2 in the rebuttal PDF for details. TLDR: CAAFE (GPT-4) + TabPFN is the strongest among all methods. Using TabPFN as a classifier, a critical difference diagram shows the statistically significant performance advantage of CAAFE to all baselines.
>
> Semantic information, i.e. the context of the dataset and its columns, is crucial and can only be captured through laborious human work or our novel approach of using LLMs - this is the core of our approach.
> To further verify and quantify this claim, we perform an experiment where the context of the dataset is left out (i.e. feature names and dataset description are not given to the LLM). We find a strong drop in performance from an average AUROC of 0.822 to 0.8 over all datasets for GPT-4. Please, see Table 1 in the rebuttal PDF for details.
>
> Why does semantic information help non-trivially?
>
> It reduces computational complexity by considering useful features. We saw CAAFE generates sophisticated features depending on up to 9 base features.
> A featurization is only validated on the training data, but performance is measured on separate test data. Overfitting can occur in the same way as it does in training a model - a featurization can be viewed as just another step in the model construction. When the number of provided samples is small or the number of tested featurizations is large, the risk of overfitting becomes larger. By considering semantically meaningful features CAAFE generates fewer but more valuable features, that thus are more likely to generalize to the test set.
>
> 2. You state that our work *"fails to catch up with the latest trends of AutoML systems"* since *"most of existing AutoML systems will perform automated feature engineering".*
>
> We are very closely following the space of AutoML and are definitely aware of AutoFE being part of many AutoML approaches. The premise of our work is that classical AutoML and AutoFE cannot capture and use contextual information, which our work seeks to do. Our original manuscript contains TabPFN which is one such AutoML method, that implicitly engineers features. For this rebuttal, we evaluated further AutoML methods, namely Autosklearn and Autogluon, and use them together with CAAFE. Here we see that while baseline feature engineering methods do not improve AutoML methods, as the improvements made by them are already captured inside, CAAFE can improve even state-of-the-art AutoML methods. We explain this because the improvements gained by CAAFE come from semantic information (as we show above), which is not captured by current AutoML methods.
>
> Please, see Table 2 in the rebuttal PDF for details. There you can see that CAAFE can improve upon AutoML methods (TabPFN, Autosklearn and Autogluon) in contrast to traditional AutoFE methods.
>
>
> 3. *"In Table 1, the increased CAAFE performance is relatively trivial for most datasets"*
>
> The improvement of CAAFE is similar to the improvement achieved by using a random forest instead of logistic regression on our datasets, which is a drastic difference. While mean AUROC is significantly affected by strong performance on tic-tac-toe, the number of wins and the ranks are not affected by outliers. We still see that CAAFE improves TabPFN on 11 out of 14 datasets, more wins than random forest has over logistic regression. It looks similar in terms of rank improvement: [Log. Reg. -> Rand. Forest: Mean rank 5.11 -> 4.44] and [TabPFN -> TabPFN + CAAFE: Mean rank 4.39 -> 3.06].
>
> ### Questions
>
> 1. *"Highly recommend the author to run the data experiments using the existing AutoML systems (listed in weakness part, they are all open source) without LLM featurizations from dataset description as baselines."*
>
> This question has been answered in part 1 of our "Weaknesses" replies.
>
>
> We hope we could address your concerns and are open to additional insights or suggestions, ensuring our research stands robust in the domain.

---

> ### Author Response · Authors · 2023-08-16
>
> Dear Reviewer QFgW,
>
> I hope this message finds you well. As the discussion period is nearing its conclusion on August 21st, we kindly request your feedback and thoughts on our recent revisions. You have given a very low score based on criticisms that we are very confident to have addressed well, providing the experiments that you requested and giving further insights into the workings of CAAFE. Please let us know if there are any additional points you'd like us to cover.
>
> Thank you for your time and consideration.

---

> > ### Comment · Reviewer_QFgW · 2023-08-21
> > **Thank you for more experiments**
> >
> > The additional experiments and evaluations are very appreciated to address my concerns and questions. I have to clarify that I'm not against the idea that semantic information will help in prediction but more of conservative in the challenges when applying in the real practice given data noises and other data uncertainties. That's why a thorough evaluation with meaningful comparison and baseline models is needed for any experiments and research like this.
> > However, the additional evaluation still cannot fully resolve my concerns. It will be better to report confidence intervals given that the difference is still trivial in some cases. Also the more experiments you run, the more careful we need to be on the multi hypothesis testing problem, that the performance improvement is random, instead of statistically significant. I adjusted my rated by two grades but unfortunately still cannot accept it confidently.

---

> > > ### Author Response · Authors · 2023-08-22
> > >
> > > Thank you very much for your reply and feedback. We will reply to your two remaining concerns about our evaluation below:
> > >
> > > *"Also the more experiments you run, the more careful we need to be on the multi-hypothesis testing problem, that the performance improvement is random, instead of statistically significant"*
> > >
> > > In our rebuttal PDF, we include Critical Difference Diagrams in Figure 3, which implement the Wilcoxon test with a Bonferroni multiple testing correction:
> > >
> > > Critical difference (CD) diagrams are a powerful tool to compare outcomes of multiple treatments over multiple observations. For instance, in machine learning research we often compare the performance of multiple methods over multiple data sets (i.e., observations).
> > >
> > > A diagram like the one above concisely represents multiple hypothesis tests that are conducted over the observed outcomes. Before anything is plotted at all, the **Friedman test** tells us whether there are significant differences at all. If this test fails, we have not sufficient data to tell any of the treatments apart and we must abort. If, however, the test sucessfully rejects this possibility we can proceed with the post-hoc analysis. In this second step, a **Wilcoxon signed-rank test** tells us whether each pair of treatments exhibits a significant difference.
> > >
> > > Since we are testing multiple hypotheses, we must adjust the Wilcoxon test with *Bonferroni's method*. For each group of treatments which we can not distinguish from the **Bonferroni-adjusted Wilcoxon test**, we add a thick line to the diagram.
> > >
> > > *"It will be better to report confidence intervals given that the difference is still trivial in some cases. "*
> > >
> > > As stated in the previous rebittal, the improvement of our method is far from trivial with larger differences than switching from logistic regression to trees for the evaluated datasets. As you state above multiple testing problems occur and we believe a statistical test or standard deviations are more appropriate to report than confidence intervals. Would you agree?

---

### Author Rebuttal · Authors · 2023-08-09

Thank you very much for your constructive feedback. We believe that the reviews have helped us to improve our work significantly. For this year's NeurIPS review, it is not possible to upload a revised paper, only a one-page rebuttal PDF. Hence, we've detailed the changes we'll be making in the final paper in our responses.

### Further evaluation of baselines

We've expanded our evaluation to include

i. AutoFE methods: FETCH and OpenFE.

ii. AutoML methods: Autosklearn and Autogluon, which perform feature engineering as part of their pipeline.

You can find detailed results in Table 2 in the rebuttal PDF, and provide a summary here:

1. We find that among all feature engineering baselines and classifiers, the strongest combination is CAAFE (GPT-4) with TabPFN in terms of mean and mean rank AUROC. In addition, we find that CAAFE is the best feature engineering method for all base classifiers in both mean and mean rank AUROC, but for logistic regression.

2. We construct a critical difference plot that performs statistical tests between the feature engineering baselines. CAAFE (GPT-4) performs statistically significantly better than the baselines when using TabPFN, the strongest base classifier. Critical difference plots perform a Wilcoxon signed rank test with correction for multiple testing.

3. Notably, CAAFE improves the performance of all methods, even AutoML methods with built-in AutoFE. This underscores CAAFE's unique ability to integrate semantic/contextual information missing in traditional feature engineering.

4. CAAFE can be combined with AutoFE baseline methods and AutoML, see Table 2 in our original manuscript. CAAFE combined with an AutoFE baseline is the strongest predictor for simple classifiers (logistic regression and random forest). Here, the strength of CAAFE (context integration) is added to the strength of the baselines (combination of a large number of features). For AutoML methods that already perform AutoFE, we do not see this effect, and only the use of CAAFE is strongest.

As stated in our work, we do not consider CAAFE to work in the same domain as the baselines. CAAFE performs the novel semantic/contextual feature engineering and obtains additional information, while baselines perform data transformation based solely on the data matrix.

### Integration of Semantic Information

1. We conducted an experiment blinding the semantic information to more explicitly show the influence of semantic information on CAAFE's performance. By excluding feature names and dataset descriptions, we found a significant drop in performance, from an average AUROC of 0.822 to 0.8. Details can be found in Table 1 of our rebuttal PDF. See Table 1 (one-page rebuttal PDF) for details.

2. Looking at the generated features in https://github.com/cafeautomatedfeatures/CAFE/tree/main/data/generated_code gives a good intuition about the value of semantic information (files with V4 were generated by GPT-4, V3 by GPT-3.5): Some of these generated features contain up to 9 variables combined in a meaningful way. Generating such combinations without a strong semantic prior of meaningful combinations would be extremely expensive. Also note that a feature is only validated on the training data, but performance is measured on separate test data. Overfitting can occur in the same way as when training a model - a feature extraction can be seen as just another step in model construction. When the number of samples provided is small, or the number of features tested is large, the risk of overfitting becomes greater. By considering semantically meaningful features, CAAFE includes a prior for features that are more likely to generalize to the test set.

---

### Decision · Program_Chairs · 2023-09-21

**Decision:**

Accept (poster)

**Comment:**

Using LLM for automating feature construction based on the available feature names is a simple and brilliant (engineering) idea.

This paper has the merit to implement this idea and conduct the experiments with i) extensive awareness of the ethical risks; ii) extensive test on datasets from OpenML and Kaggle (arguing that the latter are less prone to be viewed by LLM).

The reviewers ask for extensive additional evidence and the authors conducted experiments and comparisons with other feature construction approaches.

Although one might always ask for more experiments, the AC thinks that there is enough in the paper to give food for thought for the NeurIPS audience; detailing and discussing the methodology deployed to mitigate the ethical risks would be particularly useful.